# A Sharper Global Convergence Analysis for Average Reward Reinforcement Learning via an Actor-Critic Approach

Swetha Ganesh [1 2]    Washim Uddin Mondal [3]    Vaneet Aggarwal [1]

## Abstract

This work examines average-reward reinforcement learning with general policy parametrization. Existing state-of-the-art (SOTA) guarantees for this problem are either suboptimal or hindered by several challenges, including poor scalability with respect to the size of the state-action space, high iteration complexity, and a significant dependence on knowledge of mixing times and hitting times. To address these limitations, we propose a Multi-level Monte Carlo-based Natural Actor-Critic (MLMC-NAC) algorithm. Our work is the first to achieve a global convergence rate of $\tilde{\mathcal{O}}(1/\sqrt{T})$ for average-reward Markov Decision Processes (MDPs) (where $T$ is the horizon length), using an Actor-Critic approach. Moreover, the convergence rate does not scale with the size of the state space, therefore even being applicable to infinite state spaces.

## 1. Introduction

Reinforcement Learning (RL) is a framework where an agent interacts with a Markovian environment and maximizes the total reward it receives. The temporal dependence of the state transitions makes the problem of RL much more challenging than ordinary stochastic optimization, where data are selected in an independent and identically distributed (i.i.d.) manner. RL problems are typically analyzed via three setups: episodic, discounted reward with an infinite horizon, and average reward with an infinite horizon. The average reward framework is particularly significant for real-world applications, including robotics (Gonzalez et al., 2023), transportation (Al-Abbasi et al., 2019), communication networks (Agarwal et al., 2022), and healthcare (Tamboli et al., 2024). Model-based algorithms (Jaksch et al., 2010; Agrawal & Jia, 2017; Agarwal & Aggarwal, 2023) that learn the state transition kernel from Markovian trajectories, are well-known approaches for solving RL problems. However, these methods are typically limited to small state spaces. Policy Gradient (PG) methods, a cornerstone of RL, offer a model-free alternative that naturally supports function approximation (FA), making them well-suited for large state-action spaces. When the size of the state-action space, $SA$, is large or infinite, the framework of FA (also known as general parameterization) indexes the candidate policies by a $d$-dimensional parameter, $\theta$, where $d \ll SA$. Recently, some works have established global convergence guarantees for the average-reward setting with general policy parametrization, which we discuss below.

There are two main approaches in this context. The first is the direct PG method, where value functions are estimated directly from sampled trajectories (Bai et al., 2024; Ganesh et al., 2025b). The second is the Temporal Difference (TD)-based policy evaluation approach, commonly known as the Actor-Critic (AC) method (Patel et al., 2024; Wang et al., 2024). Currently, the order-optimal $\tilde{\mathcal{O}}(T^{-1/2})$ convergence rate result, where $T$ is the horizon length, exists for direct PG method (Ganesh et al., 2025b). However, these methods face key limitations, including poor scalability to large state-action spaces and a strong reliance on precise knowledge of mixing and hitting times to decorrelate samples, an assumption that is often impractical. In contrast, existing AC algorithms circumvent these issues but are generally more challenging to analyze. More specifically, AC methods employ a TD-based critic to estimate the value function, which helps in reducing the variance. However, this reduction comes at the cost of introducing a bias, in addition to the inherent bias arising due to Markovian sampling. Direct PG methods using Markovian sampling leverage knowledge of mixing time to obtain nearly i.i.d. samples, whereas this would not suffice for AC methods due to the additional bias from the critic. As a result, the state-of-the-art bounds for AC is $\tilde{\mathcal{O}}(1/T^{1/4})$, which is suboptimal. This raises the following question:

[1]School of Industrial Engineering, Purdue University, West Lafayette, USA [2]Department of Computer Science and Automation, Indian Institute of Science, Bengaluru, India [3]Department of Electrical Engineering, Indian Institute of Technology, Kanpur, India. Correspondence to: Vaneet Aggarwal <vaneet@purdue.edu>.

*Proceedings of the 42$^{nd}$ International Conference on Machine Learning*, Vancouver, Canada. PMLR 267, 2025. Copyright 2025 by the author(s).

| Algorithm | Infinite States and Actions? | Global Convergence | Model-Free? | Policy Parametrization |
|---|---|---|---|---|
| UCRL2 (Jaksch et al., 2010) | No | $\tilde{\mathcal{O}}(1/\sqrt{T})$ | No | Tabular |
| PSRL (Agrawal & Jia, 2017) | No | $\tilde{\mathcal{O}}(1/\sqrt{T})$ | No | Tabular |
| MDP-OOMD (Wei et al., 2020)[(1)] | No | $\tilde{\mathcal{O}}(1/\sqrt{T})$ | Yes | Tabular |
| PPGAE (Bai et al., 2024) | No | $\tilde{\mathcal{O}}(1/T^{1/4})$ | Yes | General |
| PHPG (Ganesh et al., 2025b) | No | $\tilde{\mathcal{O}}(1/\sqrt{T})$ | Yes | General |
| NAC-CFA (Wang et al., 2024) | Yes | $\tilde{\mathcal{O}}(1/T^{1/4})$ | Yes | General |
| MAC (Patel et al., 2024) | Yes | $\tilde{\mathcal{O}}(1/T^{1/4})$ | Yes | General |
| MLMC-NAC (Algorithm 1) | Yes | $\tilde{\mathcal{O}}(1/\sqrt{T})$ | Yes | General |

*Table 1.* Summary of the key results on global convergence guarantees for average reward reinforcement learning. [(1)]This work also analyzes another algorithm using the more general weakly-communicating MDP assumption while achieving a higher rate of $\tilde{\mathcal{O}}(1/T^{1/3})$.

---

*Is it possible to achieve a state-action space size independent global convergence rate of $\tilde{\mathcal{O}}\left(T^{-1/2}\right)$ for general parameterized policies in average reward infinite-horizon MDPs, using a practical, Actor-Critic approach?*

**Our Contribution:** This work answers this question affirmatively. In particular, we introduce a Multi-level Monte Carlo-based Natural Actor-Critic (MLMC-NAC) algorithm that comprises two major components. The first component, referred to as the Natural Policy Gradient (NPG) subroutine, obtains an approximate NPG direction which is used to update the policy parameter. One sub-task of obtaining the NPG is estimating the advantage function. This is done via the second component, known as the Critic subroutine that achieves its desired target via Temporal Difference (TD) learning. Both NPG and critic subroutines apply MLMC-based gradient estimators that eliminate the use of the mixing time in the algorithm. We establish that MLMC-NAC achieves a global convergence rate of $\tilde{\mathcal{O}}(T^{-1/2})$ which is optimal in the order of the horizon length $T$. The key contributions in this work are summarized as follows:

- While existing AC analyses often use relatively loose bounds, we refine the analysis to achieve sharper results. Our first step towards this is to show that the global convergence error is bounded by terms proportional to the bias and second-order error in NPG estimation (Lemma 1). Since the critic updates underlie the NPG subroutine, the NPG estimation errors are inherently linked to critic estimation errors.

- In prior AC works (Wang et al., 2024; Patel et al., 2024), the global convergence bound includes the critic error, $\mathbb{E}\|\xi_t - \xi^*\|$, where $\xi_t$ is the critic estimate at time $t$ and $\xi^*$ is the true value. Instead, using Lemma 1 and Theorem 3, our analysis refines this term to $\|\mathbb{E}[\xi_t] - \xi^*\|$, which can be significantly smaller than the previous estimate.

- Bounding $\|\mathbb{E}[\xi_t] - \xi^*\|$ still remains challenging due to Markovian noise. The critic update can be interpreted as a linear recursion with Markovian noise. Under i.i.d. noise, this term decays exponentially, but with Markovian noise,

it can remain constant (Nagaraj et al., 2020). (Nagaraj et al., 2020) mitigates this by using one sample every $t_{\text{mix}}$ steps. Instead, we leverage MLMC to reduce the bias.

- In Theorem 2, we establish a convergence rate for a generic stochastic linear recursion. Given that both the NPG and critic updates can be viewed as a stochastic linear update, this forms a basis for Theorems 3 and 4.

- Theorem 1 proves the first $\tilde{\mathcal{O}}(T^{-1/2})$ global convergence result for AC methods.

**Related works:** We will discuss the relevant works in two key areas as stated below. Our discussion primarily focuses on works that employ general parameterized policies. A summary of relevant works is available in Table 1.

**Discussion on Practicality on direct PG methods:** In direct PG methods, value function estimates are nearly unbiased but suffer from high variance, which scales with the size of the action space (Wei et al., 2020; Bai et al., 2024; Ganesh et al., 2025b). Furthermore, the convergence results depend on the hitting time, which is at least the size of the state space, making the algorithm inapplicable to large or infinite state spaces. Finally, the implementation of these algorithms require precise knowledge of mixing and hitting times to decorrelate samples, which can be impractical. In contrast, our algorithm leverages Multi-Level Monte Carlo (MLMC) to mitigate bias arising from Markovian sampling.

Other recent PG-based works include (Kumar et al., 2025), which studies the tabular policy parameterization under the assumption of exact gradient access and establishes a convergence rate of $\tilde{\mathcal{O}}(1/T)$. However, this assumption sidesteps a key challenge in PG-based methods, efficient estimation of the policy gradient, which remains a significant bottleneck in more practical settings. Another related work is (Murthy et al., 2023), which studies robust average-reward Markov decision processes (MDPs) and establishes convergence guarantees for dynamic programming approaches.

**Average Reward RL with Actor-Critic approaches:** The

authors of (Chen & Zhao, 2023; Panda & Bhatnagar, 2025; Suttle et al., 2023) provided local convergence guarantees for the AC based approaches. Recently, the global convergence for the AC methods in average reward setup have been studied in (Wang et al., 2024; Patel et al., 2024), where global sample complexity of $\tilde{\mathcal{O}}(T^{-1/4})$ is shown[1].

We note that (Suttle et al., 2023; Patel et al., 2024) uses the Multi-Level Monte Carlo (MLMC)-based AC algorithm combined with AdaGrad (Duchi et al., 2011), inspired by stochastic gradient descent (SGD) related work in (Dorfman & Levy, 2022). Unfortunately, none of these studies lead to an optimal global convergence rate which is the goal of our work. We also note that the current state-of-the-art global convergence rate for AC methods in discounted MDPs is $\mathcal{O}(T^{-1/3})$ (Xu et al., 2020; Gaur et al., 2024), and the approaches proposed in this work have the potential to be applied in that setting.

## 2. Setup

In this paper, we explore an infinite horizon reinforcement learning problem with an average reward criterion, modeled by a Markov Decision Process (MDP) represented as a tuple $\mathcal{M} = (\mathcal{S}, \mathcal{A}, r, P, \rho)$. Here $\mathcal{S}$ indicates the state space, $\mathcal{A}$ defines the action space with a size of $A$, $r : \mathcal{S} \times \mathcal{A} \to [0, 1]$ represents the reward function, $P : \mathcal{S} \times \mathcal{A} \to \Delta(\mathcal{S})$ defines the state transition function, where $\Delta(\mathcal{S})$ denotes the probability simplex over $\mathcal{S}$, and $\rho \in \Delta(\mathcal{S})$ signifies the initial distribution of states. A (stationary) policy $\pi : \mathcal{S} \to \Delta(\mathcal{A})$ determines the distribution of the action to be taken given the current state. It induces the following state transition $P^\pi : \mathcal{S} \to \Delta(\mathcal{S})$ given as $P^\pi(s, s') = \sum_{a \in \mathcal{A}} P(s'|s, a)\pi(a|s), \forall s, s' \in \mathcal{S}$. Observe that for any policy $\pi$, the sequence of states yielded by the MDP forms a Markov chain. We assume the following throughout the paper.

**Assumption 1.** The Markov chain induced by every policy $\pi$, $\{s_t\}_{t \geq 0}$, is ergodic.

Before proceeding further, we point out that we consider a parameterized class of policies $\Pi$, which consists of all policies $\pi_\theta$ such that $\theta \in \Theta$, where $\Theta \subset \mathbb{R}^d$. It is well-established that if $\mathcal{M}$ is ergodic, then $\forall \theta \in \Theta$, there exists a unique stationary $\rho$-independent distribution, denoted as $d^{\pi_\theta} \in \Delta(\mathcal{S})$, defined as follows.

$$d^{\pi_\theta}(s) = \lim_{T \to \infty} \mathbb{E}_{\pi_\theta}\left[\frac{1}{T}\sum_{t=0}^{T}\mathbf{1}(s_t = s)\bigg|s_0 \sim \rho\right] \quad (1)$$

---

[1] While (Wang et al., 2024) shows $\min_{1 \leq t \leq T}(J^* - J(\theta_t)) \leq \tilde{\mathcal{O}}(T^{-1/3})$, we note that the minimum error is strictly smaller than the average error notion that we consider in our work. Substituting the bound in Proposition 5 in their work in their average error decomposition in Step 1 of their Proof Outline yields an average error of $\tilde{\mathcal{O}}(T^{-1/4})$.

where $\mathbb{E}_{\pi_\theta}$ denotes the expectation over the distribution of all $\pi_\theta$-induced trajectories and $\mathbf{1}(\cdot)$ is an indicator function. The above distribution also obeys $(P^{\pi_\theta})^\top d^{\pi_\theta} = d^{\pi_\theta}$. With this notation in place, we define the mixing time of an MDP.

**Definition 1.** The mixing time of an MDP $\mathcal{M}$ with respect to a policy parameter $\theta$ is defined as

$$t_{\text{mix}}^\theta := \min\left\{t \geq 1 \bigg| \|(P^{\pi_\theta})^t(s, \cdot) - d^{\pi_\theta}\|_{\text{TV}} \leq \frac{1}{4}, \forall s \in \mathcal{S}\right\}$$

where $\|\cdot\|_{\text{TV}}$ denotes the total variation distance. We define $t_{\text{mix}} := \sup_{\theta \in \Theta} t_{\text{mix}}^\theta$ as the the overall mixing time. This paper assumes $t_{\text{mix}}$ to be finite.

The mixing time of an MDP measures how quickly the MDP approaches its stationary distribution when the same policy is executed repeatedly. In the average reward setting, we aim to find a policy $\pi_\theta$ that maximizes the long-term average reward defined below.

$$J^{\pi_\theta} := \lim_{T \to \infty} \mathbb{E}_{\pi_\theta}\left[\frac{1}{T}\sum_{t=0}^{T} r(s_t, a_t)\bigg|s_0 \sim \rho\right] \quad (2)$$

For simplicity, we denote $J(\theta) = J^{\pi_\theta}$. This paper uses an actor-critic approach to optimize $J$. Before proceeding further, we would like to introduce a few important terms. The action-value ($Q$) function corresponding to the policy $\pi_\theta$ is defined as

$$Q^{\pi_\theta}(s, a) = \mathbb{E}_{\pi_\theta}\left[\sum_{t=0}^{\infty}\left\{r(s_t, a_t) - J(\theta)\right\}\bigg|s_0 = s, a_0 = a\right] \quad (3)$$

We can further define the state value function as

$$V^{\pi_\theta}(s) = \mathbb{E}_{a \sim \pi_\theta(\cdot|s)}[Q^{\pi_\theta}(s, a)] \quad (4)$$

Bellman's equation, thus, takes the following form (Puterman, 2014)

$$Q^{\pi_\theta}(s, a) = r(s, a) - J(\theta) + \mathbb{E}[V^{\pi_\theta}(s')], \quad (5)$$

where the expectation is over $s' \sim P(\cdot|s, a)$. We define the advantage as $A^{\pi_\theta}(s, a) \triangleq Q^{\pi_\theta}(s, a) - V^{\pi_\theta}(s)$. With the notations in place, we express below the well-known policy gradient theorem established by (Sutton et al., 1999).

$$\nabla_\theta J(\theta) = \mathbb{E}_{(s,a) \sim \nu^{\pi_\theta}}\left[A^{\pi_\theta}(s, a)\nabla_\theta \log \pi_\theta(a|s)\right] \quad (6)$$

where $\nu^{\pi_\theta}(s, a) = d^{\pi_\theta}(s)\pi(a|s)$. Policy Gradient (PG) algorithms maximize the average reward by updating $\theta$ along the policy gradient $\nabla_\theta J(\theta)$. In contrast, Natural Policy Gradient (NPG) methods update $\theta$ along the NPG direction $\omega_\theta^*$ where

$$\omega_\theta^* = F(\theta)^\dagger \nabla_\theta J(\theta), \quad (7)$$

The symbol † denotes the Moore-Penrose pseudoinverse and $F(\theta)$ is the Fisher matrix as defined as

$$F(\theta) = \mathbb{E}_{(s,a)\sim\nu^{\pi_\theta}} \left[ \nabla_\theta \log \pi_\theta(a|s) \otimes \nabla_\theta \log \pi_\theta(a|s) \right] \quad (8)$$

where $\otimes$ symbolizes the outer product. The precoder $F(\theta)$ takes the change of the parameterized policy with respect to $\theta$ into account, thereby preventing overshooting or slow updates of $\theta$. Note that $\omega_\theta^*$ can be written as the minimizer of the function $L_{\nu^{\pi_\theta}}(\cdot, \theta)$ where

$$L_{\nu^{\pi_\theta}}(\omega, \theta) = \frac{1}{2} \mathbb{E}_{(s,a)\sim\nu^{\pi_\theta}} \left[ \left( A^{\pi_\theta}(s,a) - \omega^\top \nabla_\theta \log \pi_\theta(a|s) \right)^2 \right] \quad (9)$$

for all $\omega \in \mathbb{R}^d$. This is essentially a convex optimization that can be iteratively solved utilizing a gradient-based method. Invoking (6), one can show that

$$\nabla_\omega L_{\nu^{\pi_\theta}}(\omega, \theta) = F(\theta)\omega - \nabla_\theta J(\theta) \quad (10)$$

Note that $\nabla_\omega L_{\nu^{\pi_\theta}}(\omega, \theta)$ is not exactly computable since the transition function $P$ and hence the stationary distribution, $d^{\pi_\theta}$, and the advantage function, $A^{\pi_\theta}(\cdot, \cdot)$ are typically unknown in most practical cases. Recall that $A^{\pi_\theta}(s, a) = Q^{\pi_\theta}(s, a) - V^{\pi_\theta}(s)$. Moreover, Bellman's equation (5) states that $Q^{\pi_\theta}(s, a)$ is determined by $J(\theta)$ and $V^{\pi_\theta}$. Notice that $J(\theta)$ can be written as a solution to the following optimization problem.

$$\min_{\eta \in \mathbb{R}} R(\theta, \eta) := \frac{1}{2} \sum_{s\in\mathcal{S}} \sum_{a\in\mathcal{A}} \nu^{\pi_\theta}(s,a) \left\{ \eta - r(s,a) \right\}^2 \quad (11)$$

The above formulation allows us to compute $J(\theta)$ in a gradient-based iterative manner. In particular,

$$\nabla_\eta R(\theta, \eta) = \sum_{s\in\mathcal{S}} \sum_{a\in\mathcal{A}} \nu^{\pi_\theta}(s,a) \left\{ \eta - r(s,a) \right\} \quad (12)$$

To facilitate the estimation of the advantage function, we assume that $V^{\pi_\theta}(\cdot)$ can be approximated by a critic function $\hat{V}(\zeta_\theta, \cdot) = (\zeta_\theta)^\top \phi(\cdot)$ where $\zeta_\theta \in \mathbb{R}^m$ denotes a solution to the following optimization problem, and $\phi(s) \in \mathbb{R}^m$, $\|\phi(s)\| \le 1$ is a feature vector, $\forall s \in \mathcal{S}$.

$$\min_{\zeta \in \mathbb{R}^m} E(\theta, \zeta) := \frac{1}{2} \sum_{s\in\mathcal{S}} d^{\pi_\theta}(s) (V^{\pi_\theta}(s) - \hat{V}(\zeta, s))^2 \quad (13)$$

The above formulation paves a way to compute $\zeta_\theta$ via a gradient-based iterative procedure. Note the following.

$$\nabla_\zeta E(\theta, \zeta)$$
$$= \sum_{s\in\mathcal{S}} \sum_{a\in\mathcal{A}} \nu^{\pi_\theta}(s,a) \left\{ \zeta^\top \phi(s) - Q^{\pi_\theta}(s,a) \right\} \phi(s) \quad (14)$$

The iterative updates of $\theta$, $\eta$, and $\zeta$, along their associated gradients provided in (10), (12), and (14) form the basis of our algorithm stated next.

---

**Algorithm 1** Multi-level Monte Carlo-based Natural Actor-Critic (MLMC-NAC)

---

1: **Input:** Initial parameters $\theta_0$, $\{\omega_H^k\}$, and $\{\xi_0^k\}$, policy update stepsize $\alpha$, parameters for NPG update, $\gamma$, parameters for critic update, $\beta$, $c_\beta$, initial state $s_0 \sim \rho$, outer loop size $K$, inner loop size $H$, $T_{\max}$
2: **Initialization:** $T \leftarrow 0$
3: **for** $k = 0, 1, \cdots, K-1$ **do**
4:     **for** $h = 0, 1, \cdots, H-1$ **do** {Average reward and critic estimation}
5:         $s_0^{kh} \leftarrow s_0$, $Q_{kh} \sim \text{Geom}(1/2)$
6:         $\bar{Q}_{kh} \leftarrow 2^{Q_{kh}}$ **if** $2^{Q_{kh}} \le T_{\max}$ **else** $\bar{Q}_{kh} \leftarrow 0$
7:         **for** $t = 0, \ldots, \bar{Q}_{kh} - 1$ **do**
8:             Take action $a_t^{kh} \sim \pi_{\theta_k}(\cdot|s_t^{kh})$
9:             Collect next state $s_{t+1}^{kh} \sim P(\cdot|s_t^{kh}, a_t^{kh})$
10:            Receive reward $r(s_t^{kh}, a_t^{kh})$
11:         **end for**
12:         $T_{kh} \leftarrow \bar{Q}_{kh}$, $s_0 \leftarrow s_{T_{kh}}^{kh}$
13:         Update $\xi_h^k$ using (16) and (25)
14:         $T \leftarrow T + T_{kh}$
15:     **end for**
16:     $\xi_k \leftarrow \xi_H^k$
17:     **for** $h = H, H+1, \cdots, 2H-1$ **do** {Natural Policy Gradient (NPG) estimation}
18:         $s_0^{kh} \leftarrow s_0$, $Q_{kh} \sim \text{Geom}(1/2)$
19:         $\bar{Q}_{kh} \leftarrow 2^{Q_{kh}}$ **if** $2^{Q_{kh}} \le T_{\max}$ **else** $\bar{Q}_{kh} \leftarrow 0$
20:         **for** $t = 0, \ldots, \bar{Q}_{kh} - 1$ **do**
21:             Take action $a_t^{kh} \sim \pi_{\theta_k}(\cdot|s_t^{kh})$
22:             Collect next state $s_{t+1}^{kh} \sim P(\cdot|s_t^{kh}, a_t^{kh})$
23:            Receive reward $r(s_t^{kh}, a_t^{kh})$
24:         **end for**
25:         $T_{kh} \leftarrow \bar{Q}_{kh}$, $s_0 \leftarrow s_{T_{kh}}^{kh}$
26:         Update $\omega_h^k$ using (17) and (21)
27:         $T \leftarrow T + T_{kh}$
28:     **end for**
29:     $\omega_k \leftarrow \omega_{2H}^k$, $\theta_{k+1} \leftarrow \theta_k + \alpha\omega_k$ {Policy update}
30: **end for**

---

## 3. Proposed Algorithm

We propose a Multi-level Monte Carlo-based Natural Actor-Critic (MLMC-NAC) algorithm (Algorithm 1) that runs $K$ number of epochs (also called outer loops). The $k$th loop obtains $\xi_k = [\eta_k, \zeta_k]^\top$ where $\eta_k$ denotes an estimate of the average reward $J(\theta_k)$, and $\zeta_k$ is an estimate of the critic parameter, $\zeta_{\theta_k}$. These estimates are then used to compute the approximate NPG, $\omega_k$, which is applied to update the policy parameter $\theta_k$.

$$\theta_{k+1} = \theta_k + \alpha\omega_k \quad (15)$$

where $\alpha$ is a learning parameter. The estimate $\xi_k$ is obtained in $H$ inner loop steps. In particular, $\forall h \in \{0, \cdots, H-1\}$, we apply the following updates starting from arbitrary $\xi_0^k$,

and finally assign $\xi_H^k = \xi_k$.

$$\xi_{h+1}^k = \xi_h^k - \beta \mathbf{v}_h(\theta_k, \xi_h^k), \text{ where}$$
$$\mathbf{v}_h(\theta_k, \xi_h^k) = \left[ c_\beta \hat{\nabla}_\eta R(\theta_k, \eta_h^k), \hat{\nabla}_\zeta E(\theta_k, \xi_h^k) \right]^\top \quad (16)$$

where $c_\beta$ is a constant, and $\beta$ is a learning rate. Observe that $\hat{\nabla}_\zeta E(\theta_k, \xi_h^k)$ has dependence on $\xi_h^k = [\eta_h^k, \zeta_h^k]^\top$ due to the presence of $Q$-function in (14) while $\hat{\nabla}_\eta R(\theta_k, \eta_h^k)$ is dependent only on $\eta_h^k$ (details given later).

The approximate NPG $\omega_k$ is also obtained in $H$ inner loop steps, starting from arbitrary $\omega_H^k$, then recursively applying the stochastic gradient descent (SGD) method as stated below $\forall h \in \{H, \cdots, 2H - 1\}$, and assigning $\omega_k = \omega_{2H}^k$ [2].

$$\omega_{h+1}^k = \omega_h^k - \gamma \hat{\nabla}_\omega L_{\nu^{\pi_{\theta_k}}}(\omega_h^k, \theta_k, \xi_k) \quad (17)$$

where $\gamma$ is a learning parameter, and $\hat{\nabla}_\omega L_{\nu^{\pi_\theta}}(\omega_h^k, \theta_k, \xi_k)$ symbolizes an estimate of $\nabla_\omega L_{\nu^{\pi_{\theta_k}}}(\omega_h^k, \theta_k)$. We would like to clarify that the above gradient estimate depends on $\xi_k$ because of the presence of the advantage function in the expression of the policy gradient whose estimation needs both $\eta_k, \zeta_k$ (details given later). A process is stated below to calculate the gradient estimates used in (17) and (16).

**Gradient Estimation via MLMC:** Consider a $\pi_{\theta_k}$-induced trajectory $\mathcal{T}_{kh} = \{(s_t^{kh}, a_t^{kh}, s_{t+1}^{kh})\}_{t=0}^{l_{kh}-1}$ whose length is given as $l_{kh} = 2^{Q_{kh}}$ where $Q_{kh} \sim \text{Geom}(\frac{1}{2})$. We can write the $Q$ estimate as below $\forall t \in \{0, \cdots, l_{kh} - 1\}$ using Bellman's equation (5).

$$\hat{Q}^{\pi_{\theta_k}}(\xi_k, z_t^{kh}) = r(s_t^{kh}, a_t^{kh}) - \eta_k + \zeta_k^\top \phi(s_{t+1}^{kh}) \quad (18)$$

where $z_t^{kh} := (s_t^{kh}, a_t^{kh}, s_{t+1}^{kh})$. This leads to the advantage estimate (also called the temporal difference error) as follows $\forall t \in \{0, \cdots, l_{kh} - 1\}$.

$$\hat{A}^{\pi_{\theta_k}}(\xi_k, z_t^{kh}) = r(s_t^{kh}, a_t^{kh}) - \eta_k + \zeta_k^\top \left[ \phi(s_{t+1}^{kh}) - \phi(s_t^{kh}) \right] \quad (19)$$

Define the following quantity $\forall t \in \{0, \cdots, l_{kh} - 1\}$.

$$\mathbf{u}_t^{kh} = \hat{A}_\mathbf{u}(\theta_k, z_t^{kh})\omega_h^k - \hat{b}_\mathbf{u}(\theta_k, \xi_k, z_t^{kh}) \text{ where} \quad (20)$$
$$\hat{A}_\mathbf{u}(\theta_k, z_t^{kh}) = \nabla_\theta \log \pi_{\theta_k}(a_t^{kh}|s_t^{kh}) \otimes \nabla_\theta \log \pi_{\theta_k}(a_t^{kh}|s_t^{kh})$$
$$\hat{b}_\mathbf{u}(\theta_k, \xi_k, z_t^{kh}) = \hat{A}^{\pi_{\theta_k}}(\xi_k, z_t^{kh})\nabla_\theta \log \pi_{\theta_k}(a_t^{kh}|s_t^{kh})$$

The term $\mathbf{u}_t^{kh}$ can be thought of as a crude estimate of $\nabla_\omega L_{\nu^{\pi_{\theta_k}}}(\omega_h^k, \theta_k)$, obtained using a single transition $z_t^{kh}$. One naive way to refine this estimate is to calculate an empirical average of $\{\mathbf{u}_t^{kh}\}_{t=0}^{l_{kh}-1}$. In this work, however, we resort to the MLMC method where the final estimate is

---

[2] We initiate from $h = H$, instead of $h = 0$ to emphasize that the iteration (17) starts after $H$ iterations of (16).

given as follows.

$$\hat{\nabla}_\omega L_{\nu^{\pi_{\theta_k}}}(\omega_h^k, \theta_k, \xi_k)$$
$$= U_0 + \begin{cases} 2^Q \left( U^Q - U^{Q-1} \right), & \text{if } 2^Q \leq T_{\max} \\ 0 & \text{otherwise} \end{cases} \quad (21)$$
$$= \hat{A}_{\mathbf{u},k,h}^{\text{MLMC}}(\theta_k)\omega_h^k - \hat{b}_{\mathbf{u},k,h}^{\text{MLMC}}(\theta_k, \xi_k)$$

where $U^j = \frac{1}{2^j} \sum_{t=1}^{2^j} \mathbf{u}_t^{kh}$, $j \in \{Q - 1, Q\}$, $T_{\max} \geq 2$ is a parameter, and $\hat{A}_{\mathbf{u},k,h}^{\text{MLMC}}(\theta_k)$, and $\hat{b}_{\mathbf{u},k,h}^{\text{MLMC}}(\theta_k, \xi_k)$ denote MLMC-based estimates of samples $\{\hat{A}_\mathbf{u}(\theta_k, z_t^{kh})\}_{t=0}^{l_{kh}-1}$ and $\{\hat{b}_\mathbf{u}(\theta_k, \xi_k, z_t^{kh})\}_{t=0}^{l_{kh}-1}$ respectively.

The advantage of MLMC is that it generates the same order of bias as the empirical average of $T_{\max}$ samples but requires only $\mathcal{O}(\log T_{\max})$ samples on an average.

Using a similar method, we will now obtain an estimate of $\mathbf{v}_h(\theta_k, \xi_h^k)$. Following our earlier notations, we denote $\mathcal{T}_{kh} = \{(s_t^{kh}, a_t^{kh})\}_{t=0}^{l_{kh}-1}$ as a $\pi_{\theta_k}$-induced trajectory of length $l_{kh} = 2^{Q_{kh}}$, where $Q_{kh} \sim \text{Geom}(\frac{1}{2})$. Notice the terms stated below $\forall t \in \{0, \cdots, l_{kh} - 1\}$.

$$\mathbf{v}_t^{kh} = \begin{bmatrix} c_\beta \left\{ \eta_h^k - r(s_t^{kh}, a_t^{kh}) \right\} \\ \left\{ (\zeta_h^k)^\top \phi(s_t^{kh}) - \hat{Q}^{\pi_{\theta_k}}(\xi_h^k, z_t^{kh}) \right\} \phi(s_t^{kh}) \end{bmatrix} \quad (22)$$
$$= \hat{A}_\mathbf{v}(z_t^{kh})\xi_h^k - \hat{b}_\mathbf{v}(z_t^{kh})$$

where $z_t^{kh} := (s_t^{kh}, a_t^{kh}, s_{t+1}^{kh})$, $\hat{Q}^{\pi_{\theta_k}}(\xi_h^k, z_t^{kh})$ is given by (18) and $\hat{A}_\mathbf{v}(z_t^{kh})$, $\hat{b}_\mathbf{v}(z_t^{kh})$ are defined as

$$\hat{A}_\mathbf{v}(z_t^{kh}) = \begin{bmatrix} c_\beta & 0 \\ \phi(s_t^{kh}) & \phi(s_t^{kh}) \left[ \phi(s_t^{kh}) - \phi(s_{t+1}^{kh}) \right]^\top \end{bmatrix}, \quad (23)$$
$$\hat{b}_\mathbf{v}(z_t^{kh}) = \begin{bmatrix} c_\beta r(s_t^{kh}, a_t^{kh}) \\ r(s_t^{kh}, a_t^{kh})\phi(s_t^{kh}) \end{bmatrix} \quad (24)$$

Based on (12) and (14), the term $\mathbf{v}_t^{kh}$ can be thought of as a crude approximation of $\mathbf{v}_h(\theta_k, \xi_h^k)$ obtained using a single transition, $z_t^{kh}$. The final estimate is

$$\mathbf{v}_h(\theta_k, \xi_h^k) = V_0 + \begin{cases} 2^Q \left( V^Q - V^{Q-1} \right), & \text{if } 2^Q \leq T_{\max} \\ 0 & \text{otherwise} \end{cases}$$
$$= \hat{A}_{\mathbf{v},k,h}^{\text{MLMC}}\xi_h^k - \hat{b}_{\mathbf{v},k,h}^{\text{MLMC}} \quad (25)$$

where $V^j := 2^{-j} \sum_{t=1}^{2^j} \mathbf{v}_t^{kh}$, $j \in \{Q - 1, Q\}$. Moreover, $\hat{A}_{\mathbf{v},k,h}^{\text{MLMC}}$ and $\hat{b}_{\mathbf{v},k,h}^{\text{MLMC}}$ symbolize MLMC-based estimates of $\{\hat{A}_\mathbf{v}(z_t^{kh})\}_{t=0}^{l_{kh}-1}$ and $\{\hat{b}_\mathbf{v}(z_t^{kh})\}_{t=0}^{l_{kh}-1}$ respectively.

A few remarks are in order. Although the MLMC-based estimates achieve the same order of bias as the empirical average with a lower average sample requirement, its variance is larger. Many existing literature reduce the impact of the increased variance via AdaGrad-based parameter updates. Though such methods typically work well for general

non-convex optimization problems, it does not exploit any inherent structure of strongly convex optimization problems, thereby being sub-optimal for both the NPG-finding sub-routine and the average reward and critic updates. In this paper, we resort to a version of the standard SGD to cater to our following needs. First, by judiciously choosing the learning parameters, we prove that it is possible to achieve the optimal rate without invoking AdaGrad-type updates. Finally, although our gradient estimates suffer from bias due to the inherent error present in the critic approximation, our novel analysis suitably handles these issues.

## 4. Main Results

We first state some relevant assumptions. Let $A_{\mathbf{v}}(\theta) := \mathbb{E}_\theta\left[\hat{A}_{\mathbf{v}}(z)\right]$, and $b_{\mathbf{v}}(\theta) := \mathbb{E}_\theta\left[\hat{b}_{\mathbf{v}}(z)\right]$ where matrices $\hat{A}_{\mathbf{v}}(z)$, $\hat{b}_{\mathbf{v}}(z)$ are described in (23), (24) and the expectation $\mathbb{E}_\theta$ is computed over the distribution of $z = (s, a, s')$ where $(s, a) \sim \nu^{\pi_\theta}$, $s' \sim P(\cdot|s, a)$. For arbitrary policy parameter $\theta$, we denote $\xi_\theta^* = [A_{\mathbf{v}}(\theta)]^\dagger b_{\mathbf{v}}(\theta) = [\eta_\theta^*, \zeta_\theta^*]^\top$. Using these notations, below we state some assumptions related to the critic analysis.

**Assumption 2.** We assume the critic approximation error, $\epsilon_{\text{app}}$ (defined below) is finite.

$$\epsilon_{\text{app}} = \sup_\theta E(\theta, \zeta_\theta^*) \qquad (26)$$

**Assumption 3.** There exist $\lambda > 0$ such that $\forall \theta$

$$\mathbb{E}_\theta[\phi(s)(\phi(s) - \phi(s'))^\top] \succcurlyeq \lambda I \qquad (27)$$

where $\succcurlyeq$ is the positive semidefinite inequality and the expectation, $\mathbb{E}_\theta$ is obtained over $s \sim d^{\pi_\theta}$, $s' \sim P^{\pi_\theta}(s, \cdot)$.

Both Assumptions 2 and 3 are frequently employed in the analysis of actor-critic methods (Suttle et al., 2023; Patel et al., 2024; Wu et al., 2020; Panda & Bhatnagar, 2025). Assumption 2 intuitively relates to the quality of the feature mapping where $\epsilon_{\text{app}}$ measures the quality. Well-designed feature maps may lead to small or even zero $\epsilon_{\text{app}}$, whereas poorly designed features result in higher errors. Assumption 3 is essential for guaranteeing the uniqueness of the solution to the minimization problem (13). Assumption 3 also follows when the set of policy parameters, $\Theta$ is compact and $e \notin W_\phi$, where $e$ is the vector of all ones and $W_\phi$ is the space spanned by the feature vectors. To see this, note that if $e \notin W_\phi$, there exists $\lambda_\theta$ for every policy $\pi_\theta$ such that $\mathbb{E}_\theta[\phi(s)(\phi(s) - \phi(s'))^\top] \succcurlyeq \lambda_\theta I$ (Zhang et al., 2021b). Since $\Theta$ is compact, setting $\lambda = \inf_{\theta \in \Theta} \lambda_\theta > 0$ satisfies Assumption 3.

For large enough $c_\beta$, Assumption 3 implies that $A_{\mathbf{v}}(\theta) - (\lambda/2)I$ is positive definite (refer to Lemma 8). It also implies that $A_{\mathbf{v}}(\theta)$ is invertible. We will now state some assumptions related to the policy parameterization.

**Assumption 4.** For any $\theta$, the *transferred compatible function approximation error*, $L_{\nu^{\pi^*}}(\omega_\theta^*; \theta)$, satisfies the following inequality.

$$L_{\nu^{\pi^*}}(\omega_\theta^*; \theta) \leq \epsilon_{\text{bias}}$$

where $\omega_\theta^*$ denotes the exact NPG direction at $\theta$ defined by (7), $\pi^*$ indicates the optimal policy, and the function $L_{\nu^{\pi^*}}(\cdot, \cdot)$ is given by (9).

**Assumption 5.** For all $\theta, \theta_1, \theta_2$ and $(s, a) \in \mathcal{S} \times \mathcal{A}$, the following statements hold.

(a) $\|\nabla_\theta \log \pi_\theta(a|s)\| \leq G_1$

(b) $\|\nabla_\theta \log \pi_{\theta_1}(a|s) - \nabla_\theta \log \pi_{\theta_2}(a|s)\| \leq G_2\|\theta_1 - \theta_2\|$

**Assumption 6** (Fisher non-degenerate policy). There exists a constant $\mu > 0$ such that $F(\theta) - \mu I_d$ is positive semidefinite where $I_d$ denotes an identity matrix.

**Comments on Assumptions 4-6:** We would like to highlight that all these assumptions are commonly found in PG literature (Liu et al., 2020; Agarwal et al., 2021; Papini et al., 2018; Xu et al., 2019; Fatkhullin et al., 2023). We elaborate more on these assumptions below.

The term $\epsilon_{\text{bias}}$ captures the expressivity of the parameterized policy class. If, for example, the policy class is complete such as in the case of softmax parametrization, $\epsilon_{\text{bias}} = 0$ (Agarwal et al., 2021). However, for restricted parametrization which may not contain all stochastic policies, we have $\epsilon_{\text{bias}} > 0$. It is known that $\epsilon_{\text{bias}}$ is insignificant for rich neural parametrization (Wang et al., 2019). Note that Assumption 5 requires that the score function is bounded and Lipschitz continuous. This assumption is widely used in the analysis of PG-based methods (Liu et al., 2020; Agarwal et al., 2021; Papini et al., 2018; Xu et al., 2019; Fatkhullin et al., 2023). Assumption 6 requires that the eigenvalues of the Fisher information matrix can be bounded from below and is commonly used in obtaining global complexity bounds for PG-based procedures (Liu et al., 2020; Zhang et al., 2021a; Bai et al., 2022; Fatkhullin et al., 2023). Assumptions 5-6 were shown to hold for various examples recently including Gaussian policies with linearly parameterized means and certain neural parametrizations (Liu et al., 2020; Fatkhullin et al., 2023).

With the relevant assumptions in place, we are now ready to state our main result.

**Theorem 1.** *Consider Algorithm 1 with $K = \Theta(\sqrt{T})$, $H = \Theta(\sqrt{T}/\log(T))$. Let Assumptions 1-6 hold and $J$ be $L$-smooth. There exists a choice of parameters such that the following holds for a sufficiently large $T$.*

$$J^* - \frac{1}{K}\sum_{k=0}^{K-1} \mathbb{E}[J(\theta_k)] \leq \mathcal{O}\left(\sqrt{\epsilon_{\text{app}}} + \sqrt{\epsilon_{\text{bias}}}\right)$$

$$+ \tilde{\mathcal{O}}\left(t_{\text{mix}}^3 T^{-1/2}\right) \qquad (28)$$

*where $J^*$ is the optimal value of $J(\cdot)$.*

The values of the learning parameters used in the above theorem can be found in Appendix H. It is to be mentioned that the above bound of $\tilde{\mathcal{O}}(1/\sqrt{T})$ is a significant improvement in comparison to the state-of-the-art bounds of $\tilde{\mathcal{O}}(1/T^{1/4})$ in the average reward general parameterization setting (Bai et al., 2024; Patel et al., 2024). Also, our bounds do not depend on the size of the action space and hitting time unlike that in (Bai et al., 2024). Although (Patel et al., 2024) provides bounds with $\mathcal{O}(\sqrt{t_{\text{mix}}})$ dependence, these bounds, unfortunately, depend on the projection radius of the critic updates, $R_\omega$, which can be large and scale with $t_{\text{mix}}$ (Wei et al., 2020). In contrast, our algorithm does not use such projection operators and therefore, does not scale with $R_\omega$.

Our analysis assumes $L$-smoothness of the average reward objective $J$, a standard assumption in the PG literature. In the average-reward setting, smoothness is typically assumed, either explicitly or implicitly via Lipschitz continuity of the value function or by appealing to discounted-setting bounds (Chen & Zhao, 2023; Suttle et al., 2023; Patel et al., 2024; Ganesh et al., 2025a; Wang et al., 2024; Bai et al., 2024; Panda & Bhatnagar, 2025). A recent result (Ganesh et al., 2025b, Theorem 3) establishes a smoothness-type result under ergodicity in the average-reward, infinite-horizon setting, which could be used in our analysis but is omitted here to streamline the presentation. Algorithm 1 assumes knowledge of $L$ to set the policy learning rate, and the smoothness upper bound in the cited work depends on $t_{\text{mix}}$. However, the dependence on $t_{\text{mix}}$ in Algorithm 1 is much weaker than in existing direct policy gradient methods, which require samples to be spaced $\tilde{\mathcal{O}}(t_{\text{mix}})$ apart at each iteration.

## 5. Proof Outline

### 5.1. Policy update analysis

**Lemma 1.** *Consider any policy update rule of form*

$$\theta_{k+1} = \theta_k + \alpha \omega_k. \tag{29}$$

*If Assumptions 4 and 5 hold, then the following inequality is satisfied for any $K$.*

$$
\begin{aligned}
J^* - \frac{1}{K} \sum_{k=0}^{K-1} \mathbb{E}[J(\theta_k)] &\leq \sqrt{\epsilon_{\text{bias}}} \\
&+ \frac{G_1}{K} \sum_{k=0}^{K-1} \mathbb{E} \left\| (\mathbb{E}_k [\omega_k] - \omega_k^*) \right\| + \frac{\alpha G_2}{2K} \sum_{k=0}^{K-1} \mathbb{E} \left\| \omega_k \right\|^2 \\
&+ \frac{1}{\alpha K} \mathbb{E}_{s \sim d^{\pi^*}} [\text{KL}(\pi^*(\cdot|s) \| \pi_{\theta_0}(\cdot|s))]
\end{aligned} \tag{30}
$$

*where $\text{KL}(\cdot\|\cdot)$ is the Kullback-Leibler divergence, $\omega_k^*$ is the NPG direction $F(\theta_k)^{-1} \nabla J(\theta_k)$, $\pi^*$ is the optimal policy,*

$J^*$ is the optimal value of the function $J(\cdot)$, and $\mathbb{E}_k[\cdot]$ denotes conditional expectation given the history up to epoch $k$.

Note that the last term in (30) is $\mathcal{O}(1/K)$. The term $\mathbb{E} \left\| \omega_k \right\|^2$ can be further decomposed as

$$
\begin{aligned}
\mathbb{E} \left\| \omega_k \right\|^2 &\leq 2 \mathbb{E} \left\| \omega_k - \omega_k^* \right\|^2 + 2 \mathbb{E} \left\| \omega_k^* \right\|^2 \\
&\overset{(a)}{\leq} 2 \mathbb{E} \left\| \omega_k - \omega_k^* \right\|^2 + \frac{2}{\mu^2} \mathbb{E} \left\| \nabla_\theta J(\theta_k) \right\|^2
\end{aligned} \tag{31}
$$

where $(a)$ follows from Assumption 6 and the definition that $\omega_k^* = F(\theta_k)^{-1} \nabla_\theta J(\theta_k)$. Further, it can be proven that for the choice of $\alpha$ used in Theorem 1, we have

$$
\begin{aligned}
\frac{1}{\mu^2 K} \left( \sum_{k=0}^{K-1} \left\| \nabla_\theta J(\theta_k) \right\|^2 \right) &\leq \frac{32 L G_1^4}{\mu^4 K} \\
&+ \left( \frac{2 G_1^4}{\mu^2} + 1 \right) \left( \frac{1}{K} \sum_{k=0}^{K-1} \left\| \omega_k - \omega_k^* \right\|^2 \right)
\end{aligned} \tag{32}
$$

Evidently, one can obtain a global convergence bound by bounding the terms $\mathbb{E} \left\| \omega_k - \omega_k^* \right\|^2$, and $\mathbb{E} \left\| (\mathbb{E}_k [\omega_k] - \omega_k^*) \right\|$. These terms define the second-order error and bias of the NPG estimator, $\omega_k$. In the next subsections, we briefly describe how to obtain these bounds.

### 5.2. Analysis of a General Linear Recursion

Observe that the NPG finding subroutine (17) and the update of the critic parameter and the average reward (16) can be written in the following form for a given $k$.

$$x_{h+1} = x_h - \bar{\beta}(\hat{P}_h x_h - \hat{q}_h) \tag{33}$$

where $\hat{P}_h, \hat{q}_h$ are MLMC based estimates of the matrices $P \in \mathbb{R}^{n \times n}, q \in \mathbb{R}^n$ respectively, and $h \in \{0, \cdots, H-1\}$. Assume that the following bounds hold $\forall h$.

$$
\begin{aligned}
\mathbb{E}_h \left[ \left\| \hat{P}_h - P \right\|^2 \right] &\leq \sigma_P^2, \ \left\| \mathbb{E}_h \left[ \hat{P}_h \right] - P \right\|^2 \leq \delta_P^2, \\
\mathbb{E}_h \left[ \left\| \hat{q}_h - q \right\|^2 \right] &\leq \sigma_q^2, \ \left\| \mathbb{E}_h [\hat{q}_h] - q \right\|^2 \leq \delta_q^2, \\
\text{and} \ \left\| \mathbb{E} [\hat{q}_h] - q \right\|^2 &\leq \bar{\delta}_q^2
\end{aligned} \tag{34}
$$

where $\mathbb{E}_h$ denotes conditional expectation given history up to step $h$. Since $\mathbb{E}[\hat{q}_h] = \mathbb{E}[\mathbb{E}_h[\hat{q}_h]]$, we have $\bar{\delta}_q^2 \leq \delta_q^2$. Additionally, assume that

$$0 \prec \lambda_P I \preccurlyeq P, \ \|P\| \leq \Lambda_P \ \text{and} \ \|q\| \leq \Lambda_q \tag{35}$$

The condition that $\lambda_P > 0$ implies that $P$ is invertible. The goal of recursion (33) is to approximate the term $x^* = P^{-1} q$. We have the following result.

**Theorem 2.** *Consider the recursion* (33). *Assume that the conditions* (34), *and* (35) *hold. Also, let* $\delta_P \leq \lambda_P/8$, *and* $\bar{\beta} = \frac{2\log H}{\lambda_P H}$. *The following relation holds whenever $H$ is sufficiently large.*

$$\mathbb{E}\left[\|x_H - x^*\|^2\right] \leq \frac{\mathbb{E}\left[\|x_0 - x^*\|^2\right]}{H^2} + \tilde{\mathcal{O}}\left(\frac{R_0}{H} + R_1\right)$$

*where* $R_0 = \lambda_P^{-4}\Lambda_q^2\sigma_P^2 + \lambda_P^{-2}\sigma_q^2$, $R_1 = \lambda_P^{-2}[\delta_P^2\lambda_P^{-2}\Lambda_q^2 + \delta_q^2]$, *and* $\tilde{\mathcal{O}}(\cdot)$ *hides logarithmic factors of $H$. Moreover,*

$$\|\mathbb{E}[x_H] - x^*\|^2 \leq \frac{\|\mathbb{E}[x_0] - x^*\|^2}{H^2} + \mathcal{O}(\bar{R}_1)$$
$$+ \mathcal{O}\left(\lambda_P^{-2}\delta_P^2\left\{\mathbb{E}\left[\|x_0 - x^*\|^2\right] + \mathcal{O}\left(R_0 + R_1\right)\right\}\right)$$

*where* $\bar{R}_1 = \lambda_P^{-2}[\delta_P^2\lambda_P^{-2}\Lambda_q^2 + \bar{\delta}_q^2]$.

We shall now use Theorem 2 to characterize the estimation errors in the NPG-finding subroutine and average reward and critic updates.

### 5.3. Analysis of NPG-Finding Subroutine

In the NPG finding subroutine, the goal is to compute $\omega_k^* = [F(\theta_k)]^{-1}\nabla_\theta J(\theta_k)$. An estimate of $F(\theta_k)$ is given by $\hat{A}_{\mathbf{u},k,h}^{\text{MLMC}}(\theta_k)$, and that of the policy gradient $\nabla_\theta J(\theta_k)$ is given by $\hat{b}_{\mathbf{u},k,h}^{\text{MLMC}}(\theta_k, \xi_k)$ (see (21)). One can establish the following inequalities invoking the properties of the MLMC estimates.

**Lemma 2.** *Fix an instant $k$ of the outer loop in Algorithm 1. Given $(\theta_k, \xi_k)$, the MLMC estimates defined in* (21) *satisfy the following bounds $\forall h \in \{H, \cdots, 2H-1\}$ provided the assumptions in Theorem 1 hold.*

$(a)$ $\left\|\mathbb{E}_{k,h}\left[\hat{A}_{\mathbf{u},k,h}^{\text{MLMC}}(\theta_k)\right] - F(\theta_k)\right\|^2 \leq \mathcal{O}\left(G_1^4 t_{\text{mix}}T_{\max}^{-1}\right)$

$(b)$ $\mathbb{E}_{k,h}\left[\left\|\hat{A}_{\mathbf{u},k,h}^{\text{MLMC}}(\theta_k) - F(\theta_k)\right\|^2\right]$
$$\leq \mathcal{O}\left(G_1^4 t_{\text{mix}}\log T_{\max}\right)$$

$(c)$ $\left\|\mathbb{E}_{k,h}\left[\hat{b}_{\mathbf{u},k,h}^{\text{MLMC}}(\theta_k, \xi_k)\right] - \nabla_\theta J(\theta_k)\right\|^2$
$$\leq \mathcal{O}\left(\sigma_{\mathbf{u},k}^2 t_{\text{mix}}T_{\max}^{-1} + \delta_{\mathbf{u},k}^2\right)$$

$(d)$ $\mathbb{E}_{k,h}\left[\left\|\hat{b}_{\mathbf{u},k,h}^{\text{MLMC}}(\theta_k, \xi_k) - \nabla_\theta J(\theta_k)\right\|^2\right]$
$$\leq \mathcal{O}\left(\sigma_{\mathbf{u},k}^2 t_{\text{mix}}\log T_{\max} + \delta_{\mathbf{u},k}^2\right)$$

*where $\mathbb{E}_{k,h}$ defines the conditional expectation given the history up to the inner loop step $h$ (within the $k$th outer loop instant), $\sigma_{\mathbf{u},k}^2 = \mathcal{O}\left(G_1^2\|\xi_k\|^2\right)$ and*

$$\delta_{\mathbf{u},k}^2 = \mathcal{O}\left(G_1^2\|\xi_k - \xi_k^*\|^2 + G_1^2\epsilon_{\text{app}}\right)$$

*where* $\xi_k^* := \xi_{\theta_k}^* = [A_{\mathbf{v}}(\theta_k)]^{-1}b_{\mathbf{v}}(\theta_k)$ *and $\mathbb{E}_k$ defines the conditional expectation given the history up to the outer loop instant $k$. Moreover, given $\theta_k$, we also have*

$(e)$ $\left\|\mathbb{E}_k\left[\hat{b}_{\mathbf{u},k,h}^{\text{MLMC}}(\theta_k, \xi_k)\right] - \nabla_\theta J(\theta_k)\right\|^2$
$$\leq \mathcal{O}\left(\bar{\sigma}_{\mathbf{u},k}^2 t_{\text{mix}}T_{\max}^{-1} + \bar{\delta}_{\mathbf{u},k}^2\right)$$

*where $\bar{\sigma}_{\mathbf{u},k}^2 = \mathcal{O}(G_1^2\mathbb{E}_k\|\xi_k\|^2)$, and $\bar{\delta}_{\mathbf{u},k}^2$ is given as*

$$\bar{\delta}_{\mathbf{u},k}^2 = \mathcal{O}\left(G_1^2\|\mathbb{E}_k[\xi_k] - \xi_k^*\|^2 + G_1^2\epsilon_{\text{app}}\right)$$

Combining Lemma 2 and Theorem 2, we arrive at the following results.

**Theorem 3.** *Consider the NPG-finding recursion* (17) *with $\gamma = \frac{2\log H}{\mu H}$ and $T_{\max} = H^2$. If all assumptions in Theorem 1 hold, then for sufficiently large $c_\beta, H$*

$$\mathbb{E}_k\left[\|\omega_k - \omega_k^*\|^2\right] \leq \frac{1}{H^2}\|\omega_H^k - \omega_k^*\|^2 + \tilde{\mathcal{O}}\left(\frac{G_1^6 t_{\text{mix}}^3}{\mu^4 H}\right)$$
$$\tilde{\mathcal{O}}\left(\frac{G_1^2 c_\beta^2 t_{\text{mix}}}{\mu^2\lambda^2 H}\right) + \mu^{-2}G_1^2\mathcal{O}\left(\mathbb{E}_k\|\xi_k - \xi_k^*\|^2 + \epsilon_{\text{app}}\right)$$

*Additionally, we also have*

$$\|\mathbb{E}_k[\omega_k] - \omega_k^*\|^2 \leq \tilde{\mathcal{O}}\left(\frac{G_1^4 t_{\text{mix}}}{\mu^2 H^2}\|\omega_H^k - \omega_k^*\|^2\right)$$
$$+ \mu^{-2}G_1^2\mathcal{O}\left(\|\mathbb{E}_k[\xi_k] - \xi_k^*\|^2 + \epsilon_{\text{app}}\right)$$
$$+ \tilde{\mathcal{O}}\left(\frac{G_1^6 t_{\text{mix}}^2}{\mu^4 H^2}\mathbb{E}_k\left[\|\xi_k - \xi_k^*\|^2\right]\right)$$
$$+ \tilde{\mathcal{O}}\left(\frac{G_1^4 t_{\text{mix}}}{\mu^2 H^2}\left\{\mu^{-4}G_1^6 t_{\text{mix}}^3 + \mu^{-2}\lambda^{-2}G_1^2 c_\beta^2 t_{\text{mix}}\right\}\right)$$

### 5.4. Critic and Average Reward Analysis

The goal of the recursion (16) is to compute the term $\xi_k^* = [A_{\mathbf{v}}(\theta_k)]^{-1}b_{\mathbf{v}}(\theta_k)$. An estimate of $A_{\mathbf{v}}(\theta_k)$ is given by $\hat{A}_{\mathbf{v},k,h}^{\text{MLMC}}$ while that of $b_{\mathbf{v}}(\theta_k)$ is given by $\hat{b}_{\mathbf{v},k,h}^{\text{MLMC}}$ (see (25)). Similar to Lemma 2, we have the following result.

**Lemma 3.** *Given the parameter $\theta_k$, the MLMC estimates defined in* (25) *obey the following bounds provided the assumptions in Theorem 1 hold.*

$(a)$ $\left\|\mathbb{E}_{k,h}\left[\hat{A}_{\mathbf{v},k,h}^{\text{MLMC}}\right] - A_{\mathbf{v}}(\theta_k)\right\|^2 \leq \mathcal{O}\left(c_\beta^2 t_{\text{mix}}T_{\max}^{-1}\right)$

$(b)$ $\mathbb{E}_{k,h}\left[\left\|\hat{A}_{\mathbf{v},k,h}^{\text{MLMC}} - A_{\mathbf{v}}(\theta_k)\right\|^2\right] \leq \mathcal{O}\left(c_\beta^2 t_{\text{mix}}\log T_{\max}\right)$

$(c)$ $\left\|\mathbb{E}_{k,h}\left[\hat{b}_{\mathbf{v},k,h}^{\text{MLMC}}\right] - b_{\mathbf{v}}(\theta_k)\right\|^2 \leq \mathcal{O}\left(c_\beta^2 t_{\text{mix}}T_{\max}^{-1}\right)$

$(d)$ $\mathbb{E}_{k,h}\left[\left\|\hat{b}_{\mathbf{v},k,h}^{\text{MLMC}} - b_{\mathbf{v}}(\theta_k)\right\|^2\right] \leq \mathcal{O}\left(c_\beta^2 t_{\text{mix}}\log T_{\max}\right)$

*where $h \in \{0, 1, \cdots, H-1\}$ and $\mathbb{E}_{k,h}$ is interpreted in the same way as in Lemma 2.*

Lemma 3 and Theorem 2 lead to the following.

**Theorem 4.** *Consider the recursion* (25). *Let* $T_{\max} = H^2$, $\beta = \frac{4 \log H}{\lambda H}$. *If all assumptions of Theorem 1 hold, then the following is true for sufficiently large* $c_\beta, H$.

$$\mathbb{E}_k \left[ \|\xi_k - \xi_k^*\|^2 \right] \leq \frac{1}{H^2} \|\xi_0^k - \xi_k^*\|^2 + \tilde{\mathcal{O}} \left( \frac{c_\beta^4 t_{\mathrm{mix}}}{\lambda^4 H} \right),$$

$$\| \mathbb{E}_k[\xi_k] - \xi_k^*\|^2 \leq \mathcal{O} \left( \frac{c_\beta^2 t_{\mathrm{mix}}}{\lambda^2 H^2} \|\xi_0^k - \xi_k^*\|^2 + \frac{c_\beta^6 t_{\mathrm{mix}}^2}{\lambda^6 H^2} \right)$$

Combining Lemma 1, Theorem 3 and 4, we establish Theorem 1. We would like to emphasize the importance of the term $\bar{\delta}_q^2$ in Theorem 2. A naive analysis would have resulted in a worse upper bound in Theorem 2 that replaces $\bar{\delta}_q^2$ with $\delta_q^2$. Such degradation in Theorem 2 would have resulted in a convergence rate of $\tilde{\mathcal{O}}(T^{-1/3})$ as opposed to our current bound of $\tilde{\mathcal{O}}(T^{-1/2})$. Finally, it is to be mentioned that our convergence bound does not depend on $|\mathcal{S}|$, thereby enabling its application to large state space MDPs as long as $t_{\mathrm{mix}}$ is finite.

## 6. Conclusions

This work presents the Multi-Level Monte Carlo-based Natural Actor-Critic (MLMC-NAC) algorithm for addressing average-reward reinforcement learning challenges. The proposed method achieves an order-optimal global convergence rate of $\tilde{\mathcal{O}}(1/\sqrt{T})$, significantly surpassing the state-of-the-art results in this domain, particularly for actor-critic approaches with general policy parametrization.

Building on this line of work, Xu et al. (2025) investigated the impact of constraints. Our analysis considers a linear critic, a limitation that has been relaxed to neural critics in discounted settings (Gaur et al., 2024; Ganesh et al., 2025a). However, extending this relaxation to the average reward setting remains an open problem.

## Acknowledgement

This work was supported by the Anusandhan National Research Foundation (ANRF), India, through the Overseas Visiting Doctoral Fellowship; the Office of Naval Research under grant N00014-23-1-2532; and Cisco Systems, Inc.

## Impact Statement

The goal of this paper is to advance the current understanding of the field of Machine Learning. We do not see any potential societal consequences of our work.

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

## A. Proof of Lemma 1

The proof of the lemma is inspired by the analysis in (Mondal & Aggarwal, 2024). The major distinction is that the bound derived in (Mondal & Aggarwal, 2024) applies to the discounted reward setting, whereas our derivation pertains to the average-reward case. We begin by stating a useful lemma.

**Lemma 4** (Lemma 4, (Bai et al., 2024)). *The difference in the performance for any policies $\pi_\theta$ and $\pi_{\theta'}$ is bounded as follows*

$$J(\theta) - J(\theta') = \mathbb{E}_{s \sim d^{\pi_\theta}} \mathbb{E}_{a \sim \pi_\theta(\cdot|s)} \left[ A^{\pi_{\theta'}}(s,a) \right]. \tag{36}$$

Continuing with the proof, we have:

$$\mathbb{E}_{s \sim d^{\pi^*}} [\mathrm{KL}(\pi^*(\cdot|s) \| \pi_{\theta_k}(\cdot|s)) - \mathrm{KL}(\pi^*(\cdot|s) \| \pi_{\theta_{k+1}}(\cdot|s))]$$

$$= \mathbb{E}_{s \sim d^{\pi^*}} \mathbb{E}_{a \sim \pi^*(\cdot|s)} \left[ \log \frac{\pi_{\theta_{k+1}}(a|s)}{\pi_{\theta_k}(a|s)} \right]$$

$$\overset{(a)}{\geq} \mathbb{E}_{s \sim d^{\pi^*}} \mathbb{E}_{a \sim \pi^*(\cdot|s)} [\nabla_\theta \log \pi_{\theta_k}(a|s) \cdot (\theta_{k+1} - \theta_k)] - \frac{G_2}{2} \|\theta_{k+1} - \theta_k\|^2$$

$$= \alpha \mathbb{E}_{s \sim d^{\pi^*}} \mathbb{E}_{a \sim \pi^*(\cdot|s)} [\nabla_\theta \log \pi_{\theta_k}(a|s) \cdot \omega_k] - \frac{G_2 \alpha^2}{2} \|\omega_k\|^2$$

$$= \alpha \mathbb{E}_{s \sim d^{\pi^*}} \mathbb{E}_{a \sim \pi^*(\cdot|s)} [\nabla_\theta \log \pi_{\theta_k}(a|s) \cdot \omega_k^*] + \alpha \mathbb{E}_{s \sim d^{\pi^*}} \mathbb{E}_{a \sim \pi^*(\cdot|s)} [\nabla_\theta \log \pi_{\theta_k}(a|s) \cdot (\omega_k - \omega_k^*)] - \frac{G_2 \alpha^2}{2} \|\omega_k\|^2$$

$$= \alpha[J^* - J(\theta_k)] + \alpha \mathbb{E}_{s \sim d^{\pi^*}} \mathbb{E}_{a \sim \pi^*(\cdot|s)} [\nabla_\theta \log \pi_{\theta_k}(a|s) \cdot \omega_k^*] - \alpha[J^* - J(\theta_k)]$$

$$\qquad + \alpha \mathbb{E}_{s \sim d^{\pi^*}} \mathbb{E}_{a \sim \pi^*(\cdot|s)} [\nabla_\theta \log \pi_{\theta_k}(a|s) \cdot (\omega_k - \omega_k^*)] - \frac{G_2 \alpha^2}{2} \|\omega_k\|^2$$

$$\overset{(b)}{=} \alpha[J^* - J(\theta_k)] + \alpha \mathbb{E}_{s \sim d^{\pi^*}} \mathbb{E}_{a \sim \pi^*(\cdot|s)} \left[ \nabla_\theta \log \pi_{\theta_k}(a|s) \cdot \omega_k^* - A^{\pi_{\theta_k}}(s,a) \right]$$

$$\qquad + \alpha \mathbb{E}_{s \sim d^{\pi^*}} \mathbb{E}_{a \sim \pi^*(\cdot|s)} [\nabla_\theta \log \pi_{\theta_k}(a|s) \cdot (\omega_k - \omega_k^*)] - \frac{G_2 \alpha^2}{2} \|\omega_k\|^2$$

$$\overset{(c)}{\geq} \alpha[J^* - J(\theta_k)] - \alpha \sqrt{\mathbb{E}_{s \sim d^{\pi^*}} \mathbb{E}_{a \sim \pi^*(\cdot|s)} \left[ \left( \nabla_\theta \log \pi_{\theta_k}(a|s) \cdot \omega_k^* - A^{\pi_{\theta_k}}(s,a) \right)^2 \right]}$$

$$\qquad + \alpha \mathbb{E}_{s \sim d^{\pi^*}} \mathbb{E}_{a \sim \pi^*(\cdot|s)} [\nabla_\theta \log \pi_{\theta_k}(a|s) \cdot (\omega_k - \omega_k^*)] - \frac{G_2 \alpha^2}{2} \|\omega_k\|^2$$

$$\overset{(d)}{\geq} \alpha[J^* - J(\theta_k)] - \alpha \sqrt{\epsilon_{\mathrm{bias}}} + \alpha \mathbb{E}_{s \sim d^{\pi^*}} \mathbb{E}_{a \sim \pi^*(\cdot|s)} [\nabla_\theta \log \pi_{\theta_k}(a|s) \cdot (\omega_k - \omega_k^*)] - \frac{G_2 \alpha^2}{2} \|\omega_k\|^2.$$

Here $(a)$ and $(b)$ follow from Assumption $5(b)$ and Lemma 4, respectively. Inequality $(c)$ results from the convexity of the function $f(x) = x^2$. Lastly, $(d)$ is a consequence of Assumption 4. By taking expectations on both sides, we derive:

$$\mathbb{E} \left[ \mathbb{E}_{s \sim d^{\pi^*}} \left[ \mathrm{KL}(\pi^*(\cdot|s) \| \pi_{\theta_k}(\cdot|s)) - \mathrm{KL}(\pi^*(\cdot|s) \| \pi_{\theta_{k+1}}(\cdot|s)) \right] \right]$$

$$\geq \alpha[J^* - \mathbb{E}[J(\theta_k)]] - \alpha \sqrt{\epsilon_{\mathrm{bias}}}$$

$$\qquad + \alpha \mathbb{E} \left[ \mathbb{E}_{s \sim d^{\pi^*}} \mathbb{E}_{a \sim \pi^*(\cdot|s)} [\nabla_\theta \log \pi_{\theta_k}(a|s) \cdot (\mathbb{E}_k[\omega_k] - \omega_k^*)] \right] - \frac{G_2 \alpha^2}{2} \mathbb{E} \left[ \|\omega_k\|^2 \right]$$

$$\geq \alpha[J^* - \mathbb{E}[J(\theta_k)]] - \alpha \sqrt{\epsilon_{\mathrm{bias}}} \tag{37}$$

$$\qquad - \alpha \mathbb{E} \left[ \mathbb{E}_{s \sim d^{\pi^*}} \mathbb{E}_{a \sim \pi^*(\cdot|s)} [\|\nabla_\theta \log \pi_{\theta_k}(a|s)\| \| \mathbb{E}_k[\omega_k] - \omega_k^*\|] \right] - \frac{G_2 \alpha^2}{2} \mathbb{E} \left[ \|\omega_k\|^2 \right]$$

$$\overset{(a)}{\geq} \alpha[J^* - \mathbb{E}[J(\theta_k)]] - \alpha \sqrt{\epsilon_{\mathrm{bias}}} - \alpha G_1 \mathbb{E} \|(\mathbb{E}_k[\omega_k] - \omega_k^*)\| - \frac{G_2 \alpha^2}{2} \mathbb{E} \left[ \|\omega_k\|^2 \right]$$

where $(a)$ follows from Assumption $5(a)$. Rearranging the terms, we get,

$$J^* - \mathbb{E}[J(\theta_k)] \leq \sqrt{\epsilon_{\mathrm{bias}}} + G_1 \mathbb{E} \|(\mathbb{E}_k[\omega_k] - \omega_k^*)\| + \frac{G_2 \alpha}{2} \mathbb{E} \|\omega_k\|^2$$

$$\qquad + \frac{1}{\alpha} \mathbb{E} \left[ \mathbb{E}_{s \sim d^{\pi^*}} [\mathrm{KL}(\pi^*(\cdot|s) \| \pi_{\theta_k}(\cdot|s)) - \mathrm{KL}(\pi^*(\cdot|s) \| \pi_{\theta_{k+1}}(\cdot|s))] \right] \tag{38}$$

Adding the above inequality from $k = 0$ to $K - 1$, using the non-negativity of KL divergence, and dividing the resulting expression by $K$, we obtain the final result.

## B. Proof of Theorem 2

Let $g_h = \hat{P}_h x_h - \hat{q}_h$. To prove the first statement, observe the following relations.

$$
\begin{aligned}
\|x_{h+1} - x^*\|^2 &= \|x_h - \bar{\beta} g_h - x^*\|^2 \\
&= \|x_h - x^*\|^2 - 2\bar{\beta}\langle x_h - x^*, g_h\rangle + \bar{\beta}^2\|g_h\|^2 \\
&= \|x_h - x^*\|^2 - 2\bar{\beta}\langle x_h - x^*, P(x_h - x^*)\rangle - 2\bar{\beta}\langle x_h - x^*, g_h - P(x_h - x^*)\rangle + \bar{\beta}^2\|g_h\|^2 \\
&\overset{(a)}{\leq} \|x_h - x^*\|^2 - 2\bar{\beta}\lambda_P\|x_h - x^*\|^2 - 2\bar{\beta}\langle x_h - x^*, g_h - P(x_h - x^*)\rangle \\
&\qquad + 2\bar{\beta}^2\|g_h - P(x_h - x^*)\|^2 + 2\bar{\beta}^2\|P(x_h - x^*)\|^2 \\
&\overset{(b)}{\leq} \|x_h - x^*\|^2 - 2\bar{\beta}\lambda_P\|x_h - x^*\|^2 - 2\bar{\beta}\langle x_h - x^*, g_h - P(x_h - x^*)\rangle \\
&\qquad + 2\bar{\beta}^2\|g_h - P(x_h - x^*)\|^2 + 2\Lambda_P^2\bar{\beta}^2\|x_h - x^*\|^2
\end{aligned}
$$

where $(a)$ and $(b)$ follow from $\lambda_P I \preccurlyeq P$ and $\|P\| \leq \Lambda_P$. Taking conditional expectation $\mathbb{E}_h$ on both sides, we obtain

$$
\begin{aligned}
\mathbb{E}_h\left[\|x_{h+1} - x^*\|^2\right] &\leq (1 - 2\bar{\beta}\lambda_P + 2\Lambda_P^2\bar{\beta}^2)\|x_h - x^*\|^2 - 2\bar{\beta}\langle x_h - x^*, \mathbb{E}_h\left[g_h - P(x_h - x^*)\right]\rangle \\
&\qquad + 2\bar{\beta}^2\,\mathbb{E}_h\|g_h - P(x_h - x^*)\|^2
\end{aligned}
\tag{39}
$$

Observe that the third term in (39) can be bounded as follows.

$$
\begin{aligned}
\|g_h - P(x_h - x^*)\|^2 &= \|(\hat{P}_h - P)(x_h - x^*) + (\hat{P}_h - P)x^* + (q - \hat{q}_h)\|^2 \\
&\leq 3\|\hat{P}_h - P\|^2\|x_h - x^*\|^2 + 3\|\hat{P}_h - P\|^2\|x^*\|^2 + 3\|q - \hat{q}_h\|^2 \\
&\leq 3\|\hat{P}_h - P\|^2\|x_h - x^*\|^2 + 3\lambda_P^{-2}\Lambda_q^2\|\hat{P}_h - P\|^2 + 3\|q - \hat{q}_h\|^2
\end{aligned}
$$

where the last inequality follows from $\|x^*\|^2 = \|P^{-1}q\|^2 \leq \lambda_P^{-2}\Lambda_q^2$. Taking expectation yields

$$
\begin{aligned}
\mathbb{E}_h\|g_h - P(x_h - x^*)\|^2 &\leq 3\,\mathbb{E}_h\|\hat{P}_h - P\|^2\|x_h - x^*\|^2 + 3\lambda_P^{-2}\Lambda_q^2\,\mathbb{E}_h\|\hat{P}_h - P\|^2 + 3\,\mathbb{E}_h\|\hat{q}_h - q\|^2 \\
&\leq 3\sigma_P^2\|x_h - x^*\|^2 + 3\lambda_P^{-2}\Lambda_q^2\sigma_P^2 + 3\sigma_q^2
\end{aligned}
\tag{40}
$$

The second term in (39) can be bounded as

$$
\begin{aligned}
-\langle x_h - x^*, \mathbb{E}_h\left[g_h - P(x_h - x^*)\right]\rangle &\leq \frac{\lambda_P}{4}\|x_h - x^*\|^2 + \frac{1}{\lambda_P}\|\mathbb{E}_h[g_h - P(x_h - x^*)]\|^2 \\
&\leq \frac{\lambda_P}{4}\|x_h - x^*\|^2 + \frac{1}{\lambda_P}\left\|\left\{\mathbb{E}_h[\hat{P}_h] - P\right\}x_h + \left\{q - \mathbb{E}_h\left[\hat{q}_h\right]\right\}\right\|^2 \\
&\leq \frac{\lambda_P}{4}\|x_h - x^*\|^2 + \frac{2\delta_P^2\|x_h\|^2 + 2\delta_q^2}{\lambda_P} \\
&\leq \frac{\lambda_P}{4}\|x_h - x^*\|^2 + \frac{4\delta_P^2\|x_h - x^*\|^2 + 4\delta_P^2\lambda_P^{-2}\Lambda_q^2 + 2\delta_q^2}{\lambda_P}
\end{aligned}
\tag{41}
$$

where the last inequality follows from $\|x^*\|^2 = \|P^{-1}q\|^2 \leq \lambda_P^{-2}\Lambda_q^2$. Substituting the above bounds in (39),

$$
\begin{aligned}
\mathbb{E}_h\left[\|x_{h+1} - x^*\|^2\right] &\leq \left(1 - \frac{3\bar{\beta}\lambda_P}{2} + \frac{8\bar{\beta}\delta_P^2}{\lambda_P} + 6\bar{\beta}^2\sigma_P^2 + 2\bar{\beta}^2\Lambda_P^2\right)\|x_h - x^*\|^2 + \frac{4\bar{\beta}}{\lambda_P}\left[2\delta_P^2\lambda_P^{-2}\Lambda_q^2 + \delta_q^2\right] \\
&\qquad + 6\bar{\beta}^2\left[\lambda_P^{-2}\Lambda_q^2\sigma_P^2 + \sigma_q^2\right]
\end{aligned}
$$

For $\delta_P \leq \lambda_P/8$, and $\bar{\beta} \leq \lambda_P/[4(6\sigma_P^2 + 2\Lambda_P^2)]$, we can modify the above inequality to the following.

$$\mathbb{E}_h[\|x_{h+1} - x^*\|^2] \leq (1 - \beta\lambda_P)\|x_h - x^*\|^2 + \frac{4\bar{\beta}}{\lambda_P}\left[2\delta_P^2\lambda_P^{-2}\Lambda_q^2 + \delta_q^2\right] + 6\bar{\beta}^2\left[\lambda_P^{-2}\Lambda_q^2\sigma_P^2 + \sigma_q^2\right]$$

Taking expectation on both sides and unrolling the recursion yields

$$\mathbb{E}[\|x_H - x^*\|^2]$$

$$\leq (1 - \bar{\beta}\lambda_P)^H \mathbb{E}\|x_0 - x^*\|^2 + \sum_{h=0}^{H-1}(1 - \bar{\beta}\lambda_P)^h\left\{\frac{4\bar{\beta}}{\lambda_P}\left[2\delta_P^2\lambda_P^{-2}\Lambda_q^2 + \delta_q^2\right] + 6\bar{\beta}^2\left[\lambda_P^{-2}\Lambda_q^2\sigma_P^2 + \sigma_q^2\right]\right\}$$

$$\leq \exp\left(-H\bar{\beta}\lambda_P\right)\mathbb{E}\|x_0 - x^*\|^2 + \frac{1}{\bar{\beta}\lambda_P}\left\{\frac{4\bar{\beta}}{\lambda_P}\left[2\delta_P^2\lambda_P^{-2}\Lambda_q^2 + \delta_q^2\right] + 6\bar{\beta}^2\left[\lambda_P^{-2}\Lambda_q^2\sigma_P^2 + \sigma_q^2\right]\right\}$$

$$= \exp\left(-H\bar{\beta}\lambda_P\right)\mathbb{E}\|x_0 - x^*\|^2 + \left\{4\lambda_P^{-2}\left[2\delta_P^2\lambda_P^{-2}\Lambda_q^2 + \delta_q^2\right] + 6\bar{\beta}\lambda_P^{-1}\left[\lambda_P^{-2}\Lambda_q^2\sigma_P^2 + \sigma_q^2\right]\right\}$$

Set $\bar{\beta} = \frac{2\log H}{\lambda_P H}$ to obtain the following result.

$$\mathbb{E}\left[\|x_H - x^*\|^2\right] \leq \frac{1}{H^2}\mathbb{E}\left[\|x_0 - x^*\|^2\right] + \mathcal{O}\left(\frac{\log H}{H}\underbrace{\left\{\lambda_P^{-4}\Lambda_q^2\sigma_P^2 + \lambda_P^{-2}\sigma_q^2\right\}}_{R_0} + \underbrace{\lambda_P^{-2}\left[\delta_P^2\lambda_P^{-2}\Lambda_q^2 + \delta_q^2\right]}_{R_1}\right) \tag{42}$$

Note that, for consistency, we must have $\log H/H \leq \lambda_P^2/[8(6\sigma_P^2 + 2\Lambda_P^2)]$. To prove the second statement, observe that we have the following recursion.

$$\|\mathbb{E}[x_{h+1}] - x^*\|^2 = \|\mathbb{E}[x_h] - \bar{\beta}\mathbb{E}[g_h] - x^*\|^2$$

$$= \|\mathbb{E}[x_h] - x^*\|^2 - 2\bar{\beta}\langle\mathbb{E}[x_h] - x^*, \mathbb{E}[g_h]\rangle + \bar{\beta}^2\|\mathbb{E}[g_h]\|^2$$

$$= \|\mathbb{E}[x_h] - x^*\|^2 - 2\bar{\beta}\langle\mathbb{E}[x_h] - x^*, P(\mathbb{E}[x_h] - x^*)\rangle - 2\bar{\beta}\langle\mathbb{E}[x_h] - x^*, \mathbb{E}[g_h] - P(\mathbb{E}[x_h] - x^*)\rangle + \bar{\beta}^2\|\mathbb{E}[g_h]\|^2$$

$$\overset{(a)}{\leq} \|\mathbb{E}[x_h] - x^*\|^2 - 2\bar{\beta}\lambda_P\|\mathbb{E}[x_h] - x^*\|^2 - 2\bar{\beta}\langle\mathbb{E}[x_h] - x^*, \mathbb{E}[g_h] - P(\mathbb{E}[x_h] - x^*)\rangle$$
$$\quad + 2\bar{\beta}^2\|\mathbb{E}[g_h] - P(\mathbb{E}[x_h] - x^*)\|^2 + 2\bar{\beta}^2\|P(\mathbb{E}[x_h] - x^*)\|^2$$

$$\overset{(b)}{\leq} \|\mathbb{E}[x_h] - x^*\|^2 - 2\bar{\beta}\lambda_P\|\mathbb{E}[x_h] - x^*\|^2 - 2\bar{\beta}\langle\mathbb{E}[x_h] - x^*, \mathbb{E}[g_h] - P(\mathbb{E}[x_h] - x^*)\rangle$$
$$\quad + 2\bar{\beta}^2\|\mathbb{E}[g_h] - P(\mathbb{E}[x_h] - x^*)\|^2 + 2\Lambda_P^2\bar{\beta}^2\|\mathbb{E}[x_h] - x^*\|^2$$

$$\leq (1 - 2\bar{\beta}\lambda_P + 2\Lambda_P^2\bar{\beta}^2)\|\mathbb{E}[x_h] - x^*\|^2 - 2\bar{\beta}\langle\mathbb{E}[x_h] - x^*, \mathbb{E}[g_h] - P(\mathbb{E}[x_h] - x^*)\rangle$$
$$\quad + 2\bar{\beta}^2\|\mathbb{E}[g_h] - P(\mathbb{E}[x_h] - x^*)\|^2 \tag{43}$$

where $(a)$ and $(b)$ follow from $\lambda_P I \preccurlyeq P$ and $\|P\| \leq \Lambda_P$. The third term in the last line of (43) can be bounded as follows.

$$\|\mathbb{E}[g_h] - P(\mathbb{E}[x_h] - x^*)\|^2 = \left\|\mathbb{E}\left[(\hat{P}_h - P)(x_h - x^*)\right] + (\mathbb{E}[\hat{P}_h] - P)x^* + (q - \mathbb{E}[\hat{q}_h])\right\|^2$$

$$\leq 3\mathbb{E}\left[\|\mathbb{E}_h[\hat{P}_h] - P\|^2\|x_h - x^*\|^2\right] + 3\mathbb{E}\left[\|\mathbb{E}_h[\hat{P}_h] - P\|^2\right]\|x^*\|^2 + 3\|q - \mathbb{E}[\hat{q}_h]\|^2$$

$$\leq 3\delta_P^2\mathbb{E}\left[\|x_h - x^*\|^2\right] + 3\lambda_P^{-2}\Lambda_q^2\delta_P^2 + 3\bar{\delta}_q^2$$

$$\overset{(a)}{\leq} 3\delta_P^2\left\{\mathbb{E}\left[\|x_0 - x^*\|^2\right] + \mathcal{O}\left(R_0 + R_1\right)\right\} + 3\lambda_P^{-2}\Lambda_q^2\delta_P^2 + 3\bar{\delta}_q^2$$

where $(a)$ follows from (42). The second term in the last line of (43) can be bounded as follows.

$$-\langle\mathbb{E}[x_h] - x^*, \mathbb{E}_h\left[\mathbb{E}[g_h] - P(\mathbb{E}[x_h] - x^*)\right]\rangle$$

$$\leq \frac{\lambda_P}{4}\|\mathbb{E}[x_h] - x^*\|^2 + \frac{1}{\lambda_P}\|\mathbb{E}[g_h] - P(\mathbb{E}[x_h] - x^*)\|^2$$

$$\leq \frac{\lambda_P}{4}\|\mathbb{E}[x_h] - x^*\|^2 + \frac{3}{\lambda_P}\left[\delta_P^2\left\{\mathbb{E}\left[\|x_0 - x^*\|^2\right] + \mathcal{O}\left(R_0 + R_1\right)\right\} + \lambda_P^{-2}\Lambda_q^2\delta_P^2 + \bar{\delta}_q^2\right]$$

Substituting the above bounds in (43), we obtain the following recursion.

$$\| \mathbb{E}[x_{h+1}] - x^* \|^2 \leq \left( 1 - \frac{3\bar{\beta}\lambda_P}{2} + 2\Lambda_P^2\bar{\beta}^2 \right) \| \mathbb{E}[x_h] - x^* \|^2$$
$$+ 6\bar{\beta} \left( \bar{\beta} + \frac{1}{\lambda_P} \right) \left[ \delta_P^2 \Big\{ \mathbb{E} \left[ \|x_0 - x^*\|^2 \right] + \mathcal{O}\left( R_0 + R_1 \right) \Big\} + \lambda_P^{-2}\Lambda_q^2\delta_P^2 + \bar{\delta}_q^2 \right]$$

If $\beta < \lambda_P/(4\Lambda_P^2)$, the above bound implies the following.

$$\| \mathbb{E}[x_{h+1}] - x^* \|^2 \leq \left( 1 - \bar{\beta}\lambda_P \right) \| \mathbb{E}[x_h] - x^* \|^2$$
$$+ 6\bar{\beta} \left( \bar{\beta} + \frac{1}{\lambda_P} \right) \left[ \delta_P^2 \Big\{ \mathbb{E} \left[ \|x_0 - x^*\|^2 \right] + \mathcal{O}\left( R_0 + R_1 \right) \Big\} + \lambda_P^{-2}\Lambda_q^2\delta_P^2 + \bar{\delta}_q^2 \right]$$

Unrolling the above recursion, we obtain

$$\| \mathbb{E}[x_H] - x^* \|^2 \leq \left( 1 - \bar{\beta}\lambda_P \right)^H \| \mathbb{E}[x_0] - x^* \|^2$$
$$+ \sum_{h=0}^{H-1} 6 \left( 1 - \bar{\beta}\lambda_P \right)^h \bar{\beta} \left( \bar{\beta} + \frac{1}{\lambda_P} \right) \left[ \delta_P^2 \Big\{ \mathbb{E} \left[ \|x_0 - x^*\|^2 \right] + \mathcal{O}\left( R_0 + R_1 \right) \Big\} + \lambda_P^{-2}\Lambda_q^2\delta_P^2 + \bar{\delta}_q^2 \right]$$
$$\leq \exp\left( -H\bar{\beta}\lambda_P \right) \| \mathbb{E}[x_0] - x^* \|^2 + \frac{6}{\lambda_P} \left( \bar{\beta} + \frac{1}{\lambda_P} \right) \left[ \delta_P^2 \Big\{ \mathbb{E} \left[ \|x_0 - x^*\|^2 \right] + \mathcal{O}\left( R_0 + R_1 \right) \Big\} + \lambda_P^{-2}\Lambda_q^2\delta_P^2 + \bar{\delta}_q^2 \right]$$

Substituting $\bar{\beta} = 2\log H/(\lambda_P H)$, we finally arrive at the following result.

$$\| \mathbb{E}[x_H] - x^* \|^2 \leq \frac{1}{H^2} \| \mathbb{E}[x_0] - x^* \|^2 + 6 \left( 1 + \frac{2\log H}{H} \right) \left[ \lambda_P^{-2}\delta_P^2 \Big\{ \mathbb{E} \left[ \|x_0 - x^*\|^2 \right] + \mathcal{O}\left( R_0 + R_1 \right) \Big\} + \bar{R}_1 \right]$$

where $\bar{R}_1 = \lambda_P^{-2} \left[ \delta_P^2 \lambda_P^{-2} \Lambda_q^2 + \bar{\delta}_q^2 \right]$. This concludes the result.

## C. Properties of the MLMC Estimates

This section provides some guarantees on the error of the MLMC estimator. This is similar to the results available in (Dorfman & Levy, 2022; Suttle et al., 2023; Beznosikov et al., 2023), although (Dorfman & Levy, 2022; Beznosikov et al., 2023) consider the case of unbiased estimates while our results deal with biased estimates.

**Lemma 5.** *Consider a time-homogeneous, ergodic Markov chain $(Z_t)_{t\geq 0}$ with a unique stationary distribution $d_Z$ and a mixing time $t_{\mathrm{mix}}$. Assume that $\nabla F(x)$ is an arbitrary gradient and $\nabla F(x, Z)$ denotes an estimate of $\nabla F(x)$. Let $\| \mathbb{E}_{d_Z}[\nabla F(x, Z)] - \nabla F(x) \|^2 \leq \delta^2$ and $\| \nabla F(x, Z_t) - \mathbb{E}_{d_Z}[\nabla F(x, Z)] \|^2 \leq \sigma^2$ for all $t \geq 0$. If $Q \sim \mathrm{Geom}(1/2)$, then the following MLMC estimator*

$$g_{\mathrm{MLMC}} = g^0 + \begin{cases} 2^Q \left( g^Q - g^{Q-1} \right), & \text{if } 2^Q \leq T_{\max} \\ 0, & \text{otherwise} \end{cases} \quad \text{where } g^j = 2^{-j} \sum_{t=1}^{2^j} \nabla F(x, Z_t) \tag{44}$$

*satisfies the inequalities stated below.*

*(a) $\mathbb{E}[g_{\mathrm{MLMC}}] = \mathbb{E}[g^{\lfloor \log T_{\max} \rfloor}]$*

*(b) $\mathbb{E}[\|\nabla F(x) - g_{\mathrm{MLMC}}\|^2] \leq \mathcal{O}\left( \sigma^2 t_{\mathrm{mix}} \log_2 T_{\max} + \delta^2 \right)$*

*(c) $\|\nabla F(x) - \mathbb{E}[g_{\mathrm{MLMC}}]\|^2 \leq \mathcal{O}\left( \sigma^2 t_{\mathrm{mix}} T_{\max}^{-1} + \delta^2 \right)$*

Before proceeding to the proof, we state a useful lemma.

**Lemma 6** (Lemma 1, (Beznosikov et al., 2023))**.** *Consider the same setup as in Lemma 5. The following inequality holds.*

$$\mathbb{E} \left[ \left\| \frac{1}{N} \sum_{i=1}^N \nabla F(x, Z_i) - \mathbb{E}_{d_Z}[\nabla F(x, Z)] \right\|^2 \right] \leq \frac{C_1 t_{\mathrm{mix}}}{N} \sigma^2 \tag{45}$$

*where $N$ is a constant, $C_1 = 16(1 + \frac{1}{\ln^2 4})$, and the expectation is over the distribution of $\{Z_i\}_{i=1}^N$ emanating from any arbitrary initial distribution.*

*Proof of Lemma 5.* The statement $(a)$ can be proven as follows.

$$\mathbb{E}[g_{\mathrm{MLMC}}] = \mathbb{E}[g^0] + \sum_{j=1}^{\lfloor \log_2 T_{\max} \rfloor} \Pr\{Q = j\} \cdot 2^j \, \mathbb{E}[g^j - g^{j-1}]$$

$$= \mathbb{E}[g^0] + \sum_{j=1}^{\lfloor \log_2 T_{\max} \rfloor} \mathbb{E}[g^j - g^{j-1}] = \mathbb{E}[g^{\lfloor \log_2 T_{\max} \rfloor}]$$

For the proof of $(b)$, notice that

$$\mathbb{E}\left[\left\|\mathbb{E}_{d_z}\left[\nabla F(x, Z)\right] - g_{\mathrm{MLMC}}\right\|^2\right]$$

$$\leq 2\,\mathbb{E}\left[\left\|\mathbb{E}_{d_z}\left[\nabla F(x_t)\right] - g^0\right\|^2\right] + 2\,\mathbb{E}\left[\left\|g_{\mathrm{MLMC}} - g^0\right\|^2\right]$$

$$= 2\,\mathbb{E}\left[\left\|\mathbb{E}_{d_z}\left[\nabla F(x_t)\right] - g^0\right\|^2\right] + 2\sum_{j=1}^{\lfloor \log_2 T_{\max} \rfloor} \Pr\{Q = j\} \cdot 4^j \, \mathbb{E}\left[\left\|g^j - g^{j-1}\right\|^2\right]$$

$$= 2\,\mathbb{E}\left[\left\|\mathbb{E}_{d_z}\left[\nabla F(x_t)\right] - g^0\right\|^2\right] + 2\sum_{j=1}^{\lfloor \log_2 T_{\max} \rfloor} 2^j \, \mathbb{E}\left[\left\|g^j - g^{j-1}\right\|^2\right]$$

$$\leq 2\,\mathbb{E}\left[\left\|\mathbb{E}_{d_z}\left[\nabla F(x_t)\right] - g^0\right\|^2\right]$$

$$+ 4\sum_{j=1}^{\lfloor \log_2 T_{\max} \rfloor} 2^j \left(\mathbb{E}\left[\left\|\mathbb{E}_{d_z}\left[\nabla F(x, Z)\right] - g^{j-1}\right\|^2\right] + \mathbb{E}\left[\left\|g^j - \mathbb{E}_{d_z}\left[\nabla F(x, Z)\right]\right\|^2\right]\right)$$

$$\overset{(a)}{\leq} C_1 t_{\mathrm{mix}} \sigma^2 \left[2 + 4\sum_{j=1}^{\lfloor \log_2 T_{\max} \rfloor} 2^j \left(\frac{1}{2^{j-1}} + \frac{1}{2^j}\right)\right]$$

$$= \mathcal{O}\left(\sigma^2 t_{\mathrm{mix}} \log_2 T_{\max}\right)$$

where $(a)$ follows from Lemma 6. Using this result, we obtain the following.

$$\mathbb{E}\left[\left\|\nabla F(x) - g_{\mathrm{MLMC}}\right\|^2\right] \leq 2\,\mathbb{E}\left[\left\|\nabla F(x) - \mathbb{E}_{d_z}\left[\nabla F(x, Z)\right]\right\|^2\right] + 2\,\mathbb{E}\left[\left\|\mathbb{E}_{d_z}\left[\nabla F(x, Z)\right] - g_{\mathrm{MLMC}}\right\|^2\right]$$

$$\leq \mathcal{O}\left(\sigma^2 t_{\mathrm{mix}} \log_2 T_{\max} + \delta^2\right)$$

This completes the proof of statement $(b)$. For part $(c)$, we have

$$\left\|\nabla F(x) - \mathbb{E}\left[g_{\mathrm{MLMC}}\right]\right\|^2 \leq 2\left\|\nabla F(x) - \mathbb{E}_{d_z}\left[\nabla F(x, Z)\right]\right\|^2 + 2\left\|\mathbb{E}_{d_z}\left[\nabla F(x, Z)\right] - \mathbb{E}\left[g_{\mathrm{MLMC}}\right]\right\|^2$$

$$\leq 2\delta^2 + 2\left\|\mathbb{E}_{d_z}\left[\nabla F(x, Z)\right] - \mathbb{E}[g^{\lfloor \log_2 T_{\max} \rfloor}]\right\|^2 \overset{(a)}{\leq} 2\delta^2 + \frac{2C_1 t_{\mathrm{mix}}}{T_{\max}} \sigma^2 \tag{46}$$

where $(a)$ follows from Lemma 6. This concludes the proof of Lemma 5. $\qquad\square$

## D. Proof of Lemma 2

Fix an outer loop instant $k$ and an inner loop instant $h \in \{H, \cdots, 2H - 1\}$. Recall the definition of $\hat{A}_{\mathbf{u}}(\theta_k, \cdot)$ from (20). The following inequalities hold for any $\theta_k$ and $z_t^{kh} \in \mathcal{S} \times \mathcal{A} \times \mathcal{S}$.

$$\mathbb{E}_{\theta_k}\left[\hat{A}_{\mathbf{u}}(\theta_k, z)\right] \overset{(a)}{=} F(\theta_k), \text{ and } \left\|\hat{A}_{\mathbf{u}}(\theta_k, z_t^{kh}) - \mathbb{E}_{\theta_k}\left[\hat{A}_{\mathbf{u}}(\theta_k, z)\right]\right\|^2 \overset{(b)}{\leq} 2G_1^4$$

where $\mathbb{E}_{\theta_k}$ denotes the expectation over the distribution of $z = (s, a, s')$ where $(s, a) \sim \nu^{\pi_{\theta_k}}$, $s' \sim P(\cdot|s, a)$. The equality $(a)$ follows from the definition of the Fisher matrix, and $(b)$ is a consequence of Assumption 5. Statements (a) and (b), therefore, directly follow from Lemma 5.

To prove the other statements, recall the definition of $\hat{b}_{\mathbf{u}}(\theta_k, \xi_k, \cdot)$ from (20). Observe the following relations for arbitrary

$\theta_k, \xi_k.$

$$
\begin{aligned}
\mathbb{E}_{\theta_k}\left[\hat{b}_{\mathbf{u}}(\theta_k, \xi_k, z)\right] - \nabla_\theta J(\theta_k) &= \mathbb{E}_{\theta_k}\left[\left\{r(s,a) - \eta_k + \langle\phi(s') - \phi(s), \zeta_k\rangle\right\}\nabla_{\theta_k}\log_{\pi_{\theta_k}}(a|s)\right] - \nabla_\theta J(\theta_k) \\
&\stackrel{(a)}{=} \underbrace{\mathbb{E}_{\theta_k}\left[\left\{\eta_k^* - \eta_k + \langle\phi(s') - \phi(s), \zeta_k - \zeta_k^*\rangle\right\}\nabla_{\theta_k}\log_{\pi_{\theta_k}}(a|s)\right]}_{T_0} + \\
&\quad + \underbrace{\mathbb{E}_{\theta_k}\left[\left\{V^{\pi_{\theta_k}}(s) - \langle\phi(s), \zeta_k^*\rangle + \langle\phi(s'), \zeta_k^*\rangle - V^{\pi_{\theta_k}}(s')\right\}\nabla_{\theta_k}\log_{\pi_{\theta_k}}(a|s)\right]}_{T_1} \\
&\quad + \underbrace{\mathbb{E}_{\theta_k}\left[\left\{V^{\pi_{\theta_k}}(s') - \eta_k^* + r(s,a) - V^{\pi_{\theta_k}}(s)\right\}\nabla_{\theta_k}\log_{\pi_{\theta_k}}(a|s)\right] - \nabla_\theta J(\theta_k)}_{T_2}
\end{aligned}
$$

In $(a)$, we have used the notation that $\xi_k^* = [\eta_k^*, \zeta_k^*]^\top$. Observe that

$$
\|T_0\|^2 \stackrel{(a)}{=} \mathcal{O}\left(G_1^2 \|\xi_k - \xi_k^*\|^2\right), \quad \|T_1\|^2 \stackrel{(b)}{=} \mathcal{O}\left(G_1^2 \epsilon_{\text{app}}\right), \quad \text{and } T_2 \stackrel{(c)}{=} 0 \tag{47}
$$

where $(a)$ follows from Assumption 5 and the boundedness of the feature map, $\phi$ while $(b)$ is a consequence of Assumption 5 and 2. Finally, $(c)$ is an application of Bellman's equation. We get,

$$
\left\|\mathbb{E}_{\theta_k}\left[\hat{b}_{\mathbf{u}}(\theta_k, \xi_k, z)\right] - \nabla_\theta J(\theta_k)\right\|^2 \leq \delta_{\mathbf{u},k}^2 = \mathcal{O}\left(G_1^2\|\xi_k - \xi_k^*\|^2 + G_1^2\epsilon_{\text{app}}\right) \tag{48}
$$

Moreover, observe that, for arbitrary $z_t^{kh} \in \mathcal{S} \times \mathcal{A} \times \mathcal{S}$

$$
\left\|\hat{b}_{\mathbf{u}}(\theta_k, \xi_k, z_t^{kh}) - \mathbb{E}_{\theta_k}\left[\hat{b}_{\mathbf{u}}(\theta_k, \xi_k, z_t)\right]\right\|^2 \stackrel{(a)}{\leq} \sigma_{\mathbf{u},k}^2 = \mathcal{O}\left(G_1^2\|\xi_k\|^2\right) \tag{49}
$$

where $(a)$ follows from Assumption 5 and the boundedness of the feature map, $\phi$. We can, therefore, conclude statements (c) and (d) by applying (48) and (49) in Lemma 5. To prove the statement (e), note that

$$
\begin{aligned}
\mathbb{E}_{\theta_k}\left[\mathbb{E}_k\left[\hat{b}_{\mathbf{u}}(\theta_k, \xi_k, z)\right]\right] - \nabla_\theta J(\theta_k) &= \mathbb{E}_{\theta_k}\left[\left\{r(s,a) - \mathbb{E}_k[\eta_k] + \langle\phi(s') - \phi(s), \mathbb{E}_k[\zeta_k]\rangle\right\}\nabla_{\theta_k}\log_{\pi_{\theta_k}}(a|s)\right] - \nabla_\theta J(\theta_k) \\
&\stackrel{(a)}{=} \underbrace{\mathbb{E}_{\theta_k}\left[\left\{\eta_k^* - \mathbb{E}_k[\eta_k] + \langle\phi(s') - \phi(s), \mathbb{E}_k[\zeta_k] - \zeta_k^*\rangle\right\}\nabla_{\theta_k}\log_{\pi_{\theta_k}}(a|s)\right]}_{T_0} + \\
&\quad + \underbrace{\mathbb{E}_{\theta_k}\left[\left\{V^{\pi_{\theta_k}}(s) - \langle\phi(s), \zeta_k^*\rangle + \langle\phi(s'), \zeta_k^*\rangle - V^{\pi_{\theta_k}}(s')\right\}\nabla_{\theta_k}\log_{\pi_{\theta_k}}(a|s)\right]}_{T_1} \\
&\quad + \underbrace{\mathbb{E}_{\theta_k}\left[\left\{V^{\pi_{\theta_k}}(s') - \eta_k^* + r(s,a) - V^{\pi_{\theta_k}}(s)\right\}\nabla_{\theta_k}\log_{\pi_{\theta_k}}(a|s)\right] - \nabla_\theta J(\theta_k)}_{T_2}
\end{aligned}
$$

Observe the following bounds.

$$
\|T_0\|^2 \stackrel{(a)}{=} \mathcal{O}\left(G_1^2\|\mathbb{E}_k[\xi_k] - \xi_k^*\|^2\right), \quad \|T_1\|^2 \stackrel{(b)}{=} \mathcal{O}\left(G_1^2\epsilon_{\text{app}}\right), \quad \text{and } T_2 \stackrel{(c)}{=} 0 \tag{50}
$$

where $(a)$ follows from Assumption 5 and the boundedness of the feature map, $\phi$ while $(b)$ is a consequence of Assumption 5 and 2. Finally, $(c)$ is an application of Bellman's equation. We get,

$$
\left\|\mathbb{E}_{\theta_k}\left[\mathbb{E}_k\left[\hat{b}_{\mathbf{u}}(\theta_k, \xi_k, z)\right]\right] - \nabla_\theta J(\theta_k)\right\|^2 \leq \bar{\delta}_{\mathbf{u},k}^2 = \mathcal{O}\left(G_1^2\|\mathbb{E}_k[\xi_k] - \xi_k^*\|^2 + G_1^2\epsilon_{\text{app}}\right) \tag{51}
$$

Using the above bound, we deduce the following.

$$\left\| \mathbb{E}_k \left[ \hat{b}_{\mathbf{u},k,h}^{\mathrm{MLMC}}(\theta_k, \xi_k) \right] - \nabla_\theta J(\theta_k) \right\|^2$$

$$\leq 2 \left\| \mathbb{E}_k \left[ \hat{b}_{\mathbf{u},k,h}^{\mathrm{MLMC}}(\theta_k, \xi_k) \right] - \mathbb{E}_{\theta_k} \left[ \mathbb{E}_k \left[ \hat{b}_{\mathbf{u}}(\theta_k, \xi_k, z) \right] \right] \right\|^2 + 2 \left\| \mathbb{E}_{\theta_k} \left[ \mathbb{E}_k \left[ \hat{b}_{\mathbf{u}}(\theta_k, \xi_k, z) \right] \right] - \nabla_\theta J(\theta_k) \right\|^2$$

$$\overset{(a)}{\leq} 2 \, \mathbb{E}_k \left\| \mathbb{E}_{k,h} \left[ \hat{b}_{\mathbf{u},k,h}^{\mathrm{MLMC}}(\theta_k, \xi_k) \right] - \mathbb{E}_{\theta_k} \left[ \hat{b}_{\mathbf{u}}(\theta_k, \xi_k, z) \right] \right\|^2 + \mathcal{O} \left( \bar{\delta}_{\mathbf{u},k}^2 \right) \overset{(b)}{\leq} \mathcal{O} \left( t_{\mathrm{mix}} T_{\mathrm{max}}^{-1} \bar{\sigma}_{\mathbf{u},k}^2 + \bar{\delta}_{\mathbf{u},k}^2 \right)$$

where $(a)$ follows from (51). Moreover, $(b)$ follows from Lemma 5(a), 6, and the definition of $\bar{\sigma}_{\mathbf{u},k}^2$. This concludes the proof of Lemma 2.

## E. Proof of Lemma 3

Recall the definitions of $\hat{A}_{\mathbf{v}}(\cdot)$ and $\hat{b}_{\mathbf{v}}(\cdot)$ given in (23) and (24) respectively. Note that the following equalities hold for any $\theta_k$.

$$\mathbb{E}_{\theta_k} \left[ \hat{A}_{\mathbf{v}}(z) \right] = A_{\mathbf{v}}(\theta_k), \text{ and } \mathbb{E}_{\theta_k} \left[ \hat{b}_{\mathbf{v}}(z) \right] = b_{\mathbf{v}}(\theta_k) \tag{52}$$

where $\mathbb{E}_{\theta_k}$ denotes the expectation over the distribution of $z = (s, a, s')$ where $(s, a) \sim \nu^{\pi_{\theta_k}}$, $s' \sim P(\cdot|s, a)$. Also, for any $z = (s, a, s') \in \mathcal{S} \times \mathcal{A} \times \mathcal{S}$, we have the following.

$$\left\| \hat{A}_{\mathbf{v}}(z) \right\| \leq |c_\beta| + \|\phi(s)\| + \left\| \phi(s)(\phi(s) - \phi(s'))^\top \right\| \overset{(a)}{\leq} c_\beta + 3 = \mathcal{O}(c_\beta), \tag{53}$$

$$\left\| \hat{b}_{\mathbf{v}}(z) \right\| \leq |c_\beta r(s, a)| + \|r(s, a)\phi(s)\| \overset{(b)}{\leq} c_\beta + 1 = \mathcal{O}(c_\beta) \tag{54}$$

where $(a)$, $(b)$ hold since $|r(s, a)| \leq 1$ and $\|\phi(s)\| \leq 1$, $\forall (s, a) \in \mathcal{S} \times \mathcal{A}$. Hence, for any $z_t^{kh} \in \mathcal{S} \times \mathcal{A} \times \mathcal{S}$, we have

$$\left\| \hat{A}_{\mathbf{v}}(z_t^{kh}) - \mathbb{E}_{\theta_k} \left[ \hat{A}_{\mathbf{v}}(z) \right] \right\|^2 \leq \mathcal{O}(c_\beta^2), \text{ and } \left\| \hat{b}_{\mathbf{v}}(z_t^{kh}) - \mathbb{E}_{\theta_k} \left[ \hat{b}_{\mathbf{v}}(z) \right] \right\|^2 \leq \mathcal{O}(c_\beta^2)$$

Combining the above results with Lemma 5 establishes the result.

## F. Proof of Theorem 3

We first state an important result regarding ergodic MPDs.

**Lemma 7.** *Lemma 14, (Wei et al., 2020) For any ergodic MDP with mixing time $t_{\mathrm{mix}}$, the following holds for any policy $\pi$.*

$$|A^\pi(s, a)| = \mathcal{O}(t_{\mathrm{mix}}), \ \forall (s, a)$$

If follows from Assumptions 5, 6, and Lemma 7 that

$$\mu I \preccurlyeq F(\theta), \ \|F(\theta)\| \leq G_1^2, \text{ and } \|\nabla_\theta J(\theta)\| \leq \mathcal{O}(G_1 t_{\mathrm{mix}}) \tag{55}$$

where $\theta$ is any arbitrary policy parameter. Combining the above results with Lemma 2 and invoking Theorem 2, we arrive at the following.

$$\mathbb{E}_k \left[ \|\omega_k - \omega_k^*\|^2 \right] \leq \frac{1}{H^2} \left\| \omega_H^k - \omega_k^* \right\|^2 + \tilde{\mathcal{O}} \left( \frac{R_0}{H} + R_1 \right),$$

$$\|\mathbb{E}_k[\omega_k] - \omega_k^*\|^2 \leq \frac{1}{H^2} \left\| \omega_H^k - \omega_k^* \right\|^2 + \mathcal{O}(\bar{R}_1) + \mathcal{O} \left( \frac{G_1^4 t_{\mathrm{mix}}}{\mu^2 H^2} \left\{ \left\| \omega_H^k - \omega_k^* \right\|^2 + \tilde{\mathcal{O}}(R_0 + R_1) \right\} \right)$$

where the terms $R_0, R_1, \bar{R}_1$ are defined as follows.

$$R_0 = \tilde{\mathcal{O}} \left( \mu^{-4} G_1^6 t_{\mathrm{mix}}^3 + \mu^{-2} G_1^2 t_{\mathrm{mix}} \mathbb{E}_k \left[ \|\xi_k\|^2 \right] + \mu^{-2} G_1^2 \mathbb{E}_k \left[ \|\xi_k - \xi_k^*\|^2 \right] + \mu^{-2} G_1^2 \epsilon_{\mathrm{app}} \right),$$

$$R_1 = \mathcal{O} \left( H^{-2} \mu^{-4} G_1^6 t_{\mathrm{mix}}^3 + H^{-2} \mu^{-2} G_1^2 t_{\mathrm{mix}} \mathbb{E}_k \left[ \|\xi_k\|^2 \right] + \mu^{-2} G_1^2 \mathbb{E}_k \left[ \|\xi_k - \xi_k^*\|^2 \right] + \mu^{-2} G_1^2 \epsilon_{\mathrm{app}} \right)$$

$$\bar{R}_1 = \mathcal{O} \left( H^{-2} \mu^{-4} G_1^6 t_{\mathrm{mix}}^3 + H^{-2} \mu^{-2} G_1^2 t_{\mathrm{mix}} \mathbb{E}_k \left[ \|\xi_k\|^2 \right] + \mu^{-2} G_1^2 \|\mathbb{E}_k[\xi_k] - \xi_k^*\|^2 + \mu^{-2} G_1^2 \epsilon_{\mathrm{app}} \right)$$

Moreover, note that

$$\mathbb{E}_k \left[ \|\xi_k\|^2 \right] \leq 2\,\mathbb{E}_k \left[ \|\xi_k - \xi_k^*\|^2 \right] + 2\,\mathbb{E}_k \left[ \|\xi_k^*\|^2 \right] \overset{(a)}{\leq} \mathcal{O} \left( \mathbb{E}_k \left[ \|\xi_k - \xi_k^*\|^2 \right] + \lambda^{-2} c_\beta^2 \right)$$

where $(a)$ follows from (57) for sufficiently large $c_\beta$ and the definition that $\xi_k^* = [A_{\mathbf{v}}(\theta_k)]^{-1} b_{\mathbf{v}}(\theta_k)$. Hence,

$$\mathbb{E}_k \left[ \|\omega_k - \omega_k^*\|^2 \right] \leq \frac{1}{H^2} \left\|\omega_H^k - \omega_k^*\right\|^2 + \tilde{\mathcal{O}} \left( \frac{1}{H} \left\{ \mu^{-4} G_1^6 t_{\text{mix}}^3 + \mu^{-2}\lambda^{-2} G_1^2 c_\beta^2 t_{\text{mix}} \right\} \right)$$

$$+ \tilde{\mathcal{O}} \left( \mu^{-2} G_1^2 \, \mathbb{E}_k \left[ \|\xi_k - \xi_k^*\|^2 \right] + \mu^{-2} G_1^2 \epsilon_{\text{app}} \right),$$

$$\|\mathbb{E}_k[\omega_k] - \omega_k^*\|^2 \leq \mathcal{O} \left( \mu^{-2} G_1^2 \|\mathbb{E}_k[\xi_k] - \xi_k^*\|^2 + \mu^{-2} G_1^2 \epsilon_{\text{app}} \right)$$

$$+ \tilde{\mathcal{O}} \left( \frac{G_1^4 t_{\text{mix}}}{\mu^2 H^2} \left\{ \left\|\omega_H^k - \omega_k^*\right\|^2 + \mu^{-2} G_1^2 t_{\text{mix}} \, \mathbb{E}_k \left[ \|\xi_k - \xi_k^*\|^2 \right] + \mu^{-4} G_1^6 t_{\text{mix}}^3 + \mu^{-2}\lambda^{-2} G_1^2 c_\beta^2 t_{\text{mix}} \right\} \right)$$

This concludes the proof.

## G. Proof of Theorem 4

We start with an important result on $A_{\mathbf{v}}(\theta)$.

**Lemma 8.** *For a large enough $c_\beta$, Assumption 3 implies that $A_{\mathbf{v}}(\theta) \succeq (\lambda/2)I$ where $I$ is an identity matrix of appropriate dimension and $\theta$ is an arbitrary policy parameter.*

*Proof of Lemma 8.* Recall that $A_{\mathbf{v}}(\theta) = \mathbb{E}_\theta[\hat{A}_{\mathbf{v}}(z)]$ where $\mathbb{E}_\theta$ denotes expectation over the distribution of $z = (s, a, s')$ where $(s, a) \sim \nu^{\pi_\theta}$, $s' \sim P(\cdot|s, a)$. Hence, for any $\xi = [\eta, \zeta]$, we have

$$\xi^\top A_{\mathbf{v}}(\theta)\xi = c_\beta \eta^2 + \eta \zeta^\top \mathbb{E}_\theta \left[ \phi(s) \right] + \zeta^\top \mathbb{E}_\theta \left[ \phi(s) \left[ \phi(s) - \phi(s') \right]^\top \right] \zeta$$

$$\overset{(a)}{\geq} c_\beta \eta^2 - |\eta| \|\zeta\| + \lambda \|\zeta\|^2 \tag{56}$$

$$\geq \|\xi\|^2 \left\{ \min_{u \in [0,1]} c_\beta u - \sqrt{u(1-u)} + \lambda(1-u) \right\} \overset{(b)}{\geq} (\lambda/2) \|\xi\|^2$$

where $(a)$ is a consequence of Assumption 3 and the fact that $\|\phi(s)\| \leq 1$, $\forall s \in \mathcal{S}$. Finally, $(b)$ is satisfied when $c_\beta \geq \lambda + \sqrt{\frac{1}{\lambda^2} - 1}$. This concludes the proof of Lemma 8. $\qquad\square$

Combining Lemma 8 with (52), (53), and (54), we can, therefore, conclude that the following inequalities hold for arbitrary $\theta_k$ whenever $c_\beta \geq \lambda + \sqrt{\frac{1}{\lambda^2} - 1}$.

$$\frac{\lambda}{2} \leq \|A_{\mathbf{v}}(\theta_k)\| \leq \mathcal{O}(c_\beta), \text{ and } \|b_{\mathbf{v}}(\theta_k)\| \leq \mathcal{O}(c_\beta) \tag{57}$$

Utilizing the above result with Lemma 3 and invoking Theorem 2, we arrive at the following.

$$\mathbb{E}_k \left[ \|\xi_k - \xi_k^*\|^2 \right] \leq \frac{1}{H^2} \left\|\xi_0^k - \xi_k^*\right\|^2 + \tilde{\mathcal{O}} \left( \frac{R_0}{H} + R_1 \right),$$

$$\mathbb{E}_k \left[ \|\mathbb{E}_k[\xi_k] - \xi_k^*\|^2 \right] \leq \frac{1}{H^2} \left\|\xi_0^k - \xi_k^*\right\|^2 + \mathcal{O}(\bar{R}_1) + \mathcal{O} \left( \frac{c_\beta^2 t_{\text{mix}}}{\lambda^2 H^2} \left\{ \left\|\xi_0^k - \xi_k^*\right\|^2 + \mathcal{O}(R_0 + R_1) \right\} \right)$$

where the terms $R_0, R_1, \bar{R}_1$ are defined as follows.

$$R_0 = \tilde{\mathcal{O}} \left( \lambda^{-4} c_\beta^4 t_{\text{mix}} + \lambda^{-2} c_\beta^2 t_{\text{mix}} \right) = \tilde{\mathcal{O}} \left( \lambda^{-4} c_\beta^4 t_{\text{mix}} \right),$$

$$R_1 = \mathcal{O} \left( H^{-2} \lambda^{-4} c_\beta^4 t_{\text{mix}} + H^{-2} \lambda^{-2} c_\beta^2 t_{\text{mix}} \right) = \mathcal{O} \left( H^{-2} \lambda^{-4} c_\beta^4 t_{\text{mix}} \right)$$

$$\bar{R}_1 = \mathcal{O} \left( H^{-2} \lambda^{-4} c_\beta^4 t_{\text{mix}} + H^{-2} \lambda^{-2} c_\beta^2 t_{\text{mix}} \right) = \mathcal{O} \left( H^{-2} \lambda^{-4} c_\beta^4 t_{\text{mix}} \right)$$

Hence, we have the following results.

$$\mathbb{E}_k \left[ \|\xi_k - \xi_k^*\|^2 \right] \leq \frac{1}{H^2} \left\| \xi_0^k - \xi_k^* \right\|^2 + \tilde{\mathcal{O}} \left( \frac{c_\beta^4 t_{\text{mix}}}{\lambda^4 H} \right),$$

$$\left\| \mathbb{E}_k[\xi_k] - \xi_k^* \right\|^2 \leq \mathcal{O} \left( \frac{c_\beta^2 t_{\text{mix}}}{\lambda^2 H^2} \left\| \xi_0^k - \xi_k^* \right\|^2 \right) + \mathcal{O} \left( \frac{c_\beta^6 t_{\text{mix}}^2}{\lambda^6 H^2} \right)$$

This concludes the proof of Theorem 4.

# H. Proof of Theorem 1

Recall that the global convergence of any update of form $\theta_{k+1} = \theta_k + \alpha \omega_k$ can be bounded as

$$J^* - \frac{1}{K} \sum_{k=0}^{K-1} \mathbb{E}[J(\theta_k)] \leq \sqrt{\epsilon_{\text{bias}}} + \frac{G_1}{K} \sum_{k=0}^{K-1} \mathbb{E} \left\| (\mathbb{E}_k [\omega_k] - \omega_k^*) \right\| + \frac{\alpha G_2}{K} \sum_{k=0}^{K-1} \mathbb{E} \left\| \omega_k - \omega_k^* \right\|^2$$

$$+ \frac{\alpha \mu^{-2}}{K} \sum_{k=0}^{K-1} \mathbb{E} \left\| \nabla_\theta J(\theta_k) \right\|^2 + \frac{1}{\alpha K} \mathbb{E}_{s \sim d^{\pi^*}} \left[ \text{KL}(\pi^*(\cdot|s) \| \pi_{\theta_0}(\cdot|s)) \right].$$

(58)

We shall now derive a bound for $\frac{1}{K} \sum_{k=0}^{K-1} \|\nabla_\theta J(\theta_k)\|^2$. Given that the function $J$ is $L$-smooth, we obtain:

$$J(\theta_{k+1}) \geq J(\theta_k) + \langle \nabla_\theta J(\theta_k), \theta_{k+1} - \theta_k \rangle - \frac{L}{2} \|\theta_{k+1} - \theta_k\|^2$$

$$= J(\theta_k) + \alpha \langle \nabla_\theta J(\theta_k), \omega_k \rangle - \frac{\alpha^2 L}{2} \|\omega_k\|^2$$

$$= J(\theta_k) + \alpha \langle \nabla_\theta J(\theta_k), \omega_k^* \rangle + \alpha \langle \nabla_\theta J(\theta_k), \omega_k - \omega_k^* \rangle - \frac{\alpha^2 L}{2} \|\omega_k - \omega_k^* + \omega_k^*\|^2$$

$$\overset{(a)}{\geq} J(\theta_k) + \alpha \left\langle \nabla_\theta J(\theta_k), F(\theta_k)^{-1} \nabla_\theta J(\theta_k) \right\rangle + \alpha \langle \nabla_\theta J(\theta_k), \omega_k - \omega_k^* \rangle$$
$$- \alpha^2 L \|\omega_k - \omega_k^*\|^2 - \alpha^2 L \|\omega_k^*\|^2$$

$$\overset{(b)}{\geq} J(\theta_k) + \frac{\alpha}{G_1^2} \|\nabla_\theta J(\theta_k)\|^2 + \alpha \langle \nabla_\theta J(\theta_k), \omega_k - \omega_k^* \rangle - \alpha^2 L \|\omega_k - \omega_k^*\|^2 - \alpha^2 L \|\omega_k^*\|^2$$

$$= J(\theta_k) + \frac{\alpha}{2G_1^2} \|\nabla_\theta J(\theta_k)\|^2 + \frac{\alpha}{2G_1^2} \left[ \|\nabla_\theta J(\theta_k)\|^2 + 2G_1^2 \langle \nabla_\theta J(\theta_k), \omega_k - \omega_k^* \rangle + G_1^4 \|\omega_k - \omega_k^*\|^2 \right]$$
$$- \left( \frac{\alpha G_1^2}{2} + \alpha^2 L \right) \|\omega_k - \omega_k^*\|^2 - \alpha^2 L \|\omega_k^*\|^2$$

(59)

$$= J(\theta_k) + \frac{\alpha}{2G_1^2} \|\nabla_\theta J(\theta_k)\|^2 + \frac{\alpha}{2G_1^2} \|\nabla_\theta J(\theta_k) + G_1^2(\omega_k - \omega_k^*)\|^2 - \left( \frac{\alpha G_1^2}{2} + \alpha^2 L \right) \|\omega_k - \omega_k^*\|^2$$
$$- \alpha^2 L \|\omega_k^*\|^2$$

$$\geq J(\theta_k) + \frac{\alpha}{2G_1^2} \|\nabla_\theta J(\theta_k)\|^2 - \left( \frac{\alpha G_1^2}{2} + \alpha^2 L \right) \|\omega_k - \omega_k^*\|^2 - \alpha^2 L \|F(\theta_k)^{-1} \nabla_\theta J(\theta_k)\|^2$$

$$\overset{(c)}{\geq} J(\theta_k) + \left( \frac{\alpha}{2G_1^2} - \frac{\alpha^2 L}{\mu^2} \right) \|\nabla_\theta J(\theta_k)\|^2 - \left( \frac{\alpha G_1^2}{2} + \alpha^2 L \right) \|\omega_k - \omega_k^*\|^2$$

where $(a)$ use the Cauchy-Schwarz inequality and the definition that $\omega_k^* = F(\theta_k)^{-1} \nabla_\theta J(\theta_k)$. Relations $(b)$, and $(c)$ are consequences of Assumption 5(a) and 6 respectively. Summing the above inequality over $k \in \{0, \cdots, K-1\}$, rearranging

the terms and substituting $\alpha = \frac{\mu^2}{4G_1^2 L}$, we obtain

$$
\begin{aligned}
\frac{\mu^2}{16 G_1^4 L}\left(\frac{1}{K}\sum_{k=0}^{K-1}\|\nabla_\theta J(\theta_k)\|^2\right) &\leq \frac{J(\theta_K)-J(\theta_0)}{K}+\left(\frac{\mu^2}{8L}+\frac{\mu^4}{16 G_1^4 L}\right)\left(\frac{1}{K}\sum_{k=0}^{K-1}\|\omega_k-\omega_k^*\|^2\right) \\
&\overset{(a)}{\leq}\frac{2}{K}+\left(\frac{\mu^2}{8L}+\frac{\mu^4}{16 G_1^4 L}\right)\left(\frac{1}{K}\sum_{k=0}^{K-1}\|\omega_k-\omega_k^*\|^2\right)
\end{aligned}
\tag{60}
$$

where $(a)$ uses the fact that $J(\cdot)$ is absolutely bounded above by 1. Inequality (60) can be simplified as follows.

$$
\frac{\mu^{-2}}{K}\left(\sum_{k=0}^{K-1}\|\nabla_\theta J(\theta_k)\|^2\right)\leq \frac{32 L G_1^4}{\mu^4 K}+\left(\frac{2 G_1^4}{\mu^2}+1\right)\left(\frac{1}{K}\sum_{k=0}^{K-1}\|\omega_k-\omega_k^*\|^2\right)
\tag{61}
$$

Now all that is left is to bound $\mathbb{E}\left[\|\omega_k-\omega_k^*\|^2\right]$ and $\|\mathbb{E}_k[\omega_k]-\omega_k^*\|$. Assume $\xi_0^k = 0, \forall k$. From Theorem 4, we have

$$
\mathbb{E}_k\left[\|\xi_k-\xi_k^*\|^2\right]\leq \frac{1}{H^2}\|\xi_k^*\|^2+\tilde{\mathcal{O}}\left(\frac{c_\beta^4 t_{\mathrm{mix}}}{\lambda^4 H}\right)\overset{(a)}{=}\tilde{\mathcal{O}}\left(\frac{c_\beta^4 t_{\mathrm{mix}}}{\lambda^4 H}\right),
\tag{62}
$$

$$
\|\mathbb{E}_k[\xi_k]-\xi_k^*\|^2\leq \mathcal{O}\left(\frac{c_\beta^2 t_{\mathrm{mix}}}{\lambda^2 H^2}\|\xi_k^*\|^2\right)+\mathcal{O}\left(\frac{c_\beta^6 t_{\mathrm{mix}}^2}{\lambda^6 H^2}\right)\overset{(b)}{=}\mathcal{O}\left(\frac{c_\beta^6 t_{\mathrm{mix}}^2}{\lambda^6 H^2}\right)
\tag{63}
$$

The relations $(a)$, $(b)$ are due to the fact that $\|\xi_k^*\|^2 = \left\|[A_{\mathbf{v}}(\theta_k)]^{-1}b_{\mathbf{v}}(\theta_k)\right\|^2\leq \mathcal{O}\left(\lambda^{-2}c_\beta^2\right)$ where the last inequality is a consequence of (57). Assume $\omega_H^k = 0, \forall k$. We have the following from Theorem 3.

$$
\begin{aligned}
\mathbb{E}_k\left[\|\omega_k-\omega_k^*\|^2\right] &\leq \frac{1}{H^2}\|\omega_k^*\|^2+\tilde{\mathcal{O}}\left(\frac{1}{H}\left\{\mu^{-4}G_1^6 t_{\mathrm{mix}}^3+\mu^{-2}\lambda^{-2}G_1^2 c_\beta^2 t_{\mathrm{mix}}\right\}\right)+\mu^{-2}G_1^2\tilde{\mathcal{O}}\left(\mathbb{E}_k\left[\|\xi_k-\xi_k^*\|^2\right]+\epsilon_{\mathrm{app}}\right) \\
&\overset{(a)}{\leq}\mathcal{O}\left(\frac{G_1^2 t_{\mathrm{mix}}^2}{\mu^2 H^2}\right)+\tilde{\mathcal{O}}\left(\frac{1}{H}\left\{\mu^{-4}G_1^6 t_{\mathrm{mix}}^3+\mu^{-2}\lambda^{-2}G_1^2 c_\beta^2 t_{\mathrm{mix}}\right\}\right)+\frac{G_1^2}{\mu^2}\tilde{\mathcal{O}}\left(\frac{c_\beta^4 t_{\mathrm{mix}}}{\lambda^4 H}+\epsilon_{\mathrm{app}}\right) \\
&\overset{(b)}{\leq}\tilde{\mathcal{O}}\left(\frac{1}{H}\left\{\mu^{-4}G_1^6 t_{\mathrm{mix}}^3+\mu^{-2}\lambda^{-4}G_1^2 c_\beta^4 t_{\mathrm{mix}}\right\}\right)+\frac{G_1^2}{\mu^2}\mathcal{O}\left(\epsilon_{\mathrm{app}}\right)
\end{aligned}
\tag{64}
$$

Inequality $(a)$ utilizes the fact that $\|\omega_k^*\|^2 = \left\|F(\theta_k)^\dagger\nabla_\theta J(\theta_k)\right\|^2\leq \mathcal{O}\left(\mu^{-2}G_1^2 t_{\mathrm{mix}}^2\right)$ where the last inequality follows from Assumption 5, 6, and Lemma 7. We also apply (62) to prove $(a)$ whereas $(b)$ is established by retaining only the dominant terms. Theorem 3 also states that

$$
\begin{aligned}
\|\mathbb{E}_k[\omega_k]-\omega_k^*\|^2 &\leq \mathcal{O}\left(\mu^{-2}G_1^2\|\mathbb{E}_k[\xi_k]-\xi_k^*\|^2+\mu^{-2}G_1^2\epsilon_{\mathrm{app}}\right) \\
&\quad +\mathcal{O}\left(\frac{G_1^4 t_{\mathrm{mix}}}{\mu^2 H^2}\left\{\|\omega_k^*\|^2+\mu^{-2}G_1^2 t_{\mathrm{mix}}\mathbb{E}_k\left[\|\xi_k-\xi_k^*\|^2\right]+\mu^{-4}G_1^6 t_{\mathrm{mix}}^3+\mu^{-2}\lambda^{-2}G_1^2 c_\beta^2 t_{\mathrm{mix}}\right\}\right) \\
&\overset{(a)}{\leq}\tilde{\mathcal{O}}\left(\frac{G_1^2 c_\beta^6 t_{\mathrm{mix}}^2}{\mu^2\lambda^6 H^2}+\frac{G_1^2}{\mu^2}\epsilon_{\mathrm{app}}\right)+\tilde{\mathcal{O}}\left(\frac{1}{H^2}\left\{\mu^{-4}G_1^6 t_{\mathrm{mix}}^3+\frac{G_1^6 c_\beta^4 t_{\mathrm{mix}}^3}{\mu^4\lambda^4 H}+\mu^{-6}G_1^{10}t_{\mathrm{mix}}^4+\mu^{-4}\lambda^{-2}G_1^6 c_\beta^2 t_{\mathrm{mix}}^2\right\}\right) \\
&\overset{(b)}{\leq}\tilde{\mathcal{O}}\left(\frac{1}{H^2}\left\{\mu^{-6}G_1^{10}t_{\mathrm{mix}}^4+\mu^{-2}\lambda^{-6}G_1^2 c_\beta^6 t_{\mathrm{mix}}^2+\mu^{-4}\lambda^{-2}G_1^6 c_\beta^2 t_{\mathrm{mix}}^2\right\}\right)+\frac{G_1^2}{\mu^2}\mathcal{O}\left(\epsilon_{\mathrm{app}}\right)
\end{aligned}
\tag{65}
$$

where $(a)$ is a consequence of (62), (63), and the upper bound $\|\omega_k^*\|^2\leq \mathcal{O}\left(\mu^{-2}G_1^2 t_{\mathrm{mix}}^2\right)$ derived earlier. Inequality $(b)$ retains only the dominant terms. Combining (58), (61), (64), and (65), we arrive at the following.

$$
\begin{aligned}
J^*-\frac{1}{K}\sum_{k=0}^{K-1}\mathbb{E}[J(\theta_k)] &\leq \sqrt{\epsilon_{\mathrm{bias}}}+\tilde{\mathcal{O}}\left(\frac{1}{H}\left\{\mu^{-3}G_1^6 t_{\mathrm{mix}}^2+\mu^{-1}\lambda^{-3}G_1^2 c_\beta^3 t_{\mathrm{mix}}+\mu^{-2}\lambda^{-1}G_1^4 c_\beta t_{\mathrm{mix}}\right\}\right)+\frac{G_1^2}{\mu}\mathcal{O}\left(\sqrt{\epsilon_{\mathrm{app}}}\right) \\
&\quad +\frac{1}{L}\left(G_1^2+\frac{\mu^2 G_2}{G_1^2}\right)\left[\tilde{\mathcal{O}}\left(\frac{1}{H}\left\{\mu^{-4}G_1^6 t_{\mathrm{mix}}^3+\mu^{-2}\lambda^{-4}G_1^2 c_\beta^4 t_{\mathrm{mix}}\right\}\right)+\frac{G_1^2}{\mu^2}\mathcal{O}\left(\epsilon_{\mathrm{app}}\right)\right]+\mathcal{O}\left(\frac{G_1^2 L}{\mu^2 K}\right)
\end{aligned}
\tag{66}
$$

We get the desired result by substituting the values of $H, K$ as stated in the theorem. We want to emphasize that the $G_1^2$ factor with the $\sqrt{\epsilon_{\text{app}}}$ term is a standard component in actor-critic results with a linear critic (Suttle et al., 2023; Patel et al., 2024). However, this factor is often not explicitly mentioned in previous works, whereas we have included it here.

