# OpenReview forum: "A Sharper Global Convergence Analysis for Average Reward Reinforcement Learning via an Actor-Critic Approach"
_ICML.cc/2025/Conference — ICML 2025 poster_

### Official Review · Reviewer_en8p · 2025-02-20

**Overall Recommendation:** 3

**Summary:**

This paper investigates average-reward reinforcement learning with general policy parametrization. The authors propose a Multi-level Monte Carlo-based Natural Actor-Critic (MLMC-NAC) algorithm and provide a finite-sample analysis.

Update after rebuttal:

The authors have satisfactorily addressed all of my concerns in the rebuttal, including the clarification regarding the single-loop versus double-loop structure. Based on their responses, I am increasing my score to 3.

**Claims And Evidence:**

Yes

**Essential References Not Discussed:**

No

**Experimental Designs Or Analyses:**

N/A

**Methods And Evaluation Criteria:**

Yes

**Other Comments Or Suggestions:**

N/A

**Other Strengths And Weaknesses:**

Strengths:
The authors propose a Multi-level Monte Carlo-based Natural Actor-Critic algorithm that achieves a global convergence rate of $\tilde{O}(1/\sqrt{T})$. The algorithm eliminates the need for knowledge of mixing times. In addition, the finite-sample analysis provided in the paper extends to infinite state and action spaces.

Weaknesses:
Please refer to Question section.

**Questions For Authors:**

My main concern is about the high-level idea. Typically, addressing Markovian noise requires either a projection step or the assumption of a finite action space. However, it is unclear to me how the authors handle Markovian noise without either of these approaches. Could the authors provide a high-level explanation of their methodology?

Besides, some other concerns are:

1. To establish Theorem 1, the objective function $J$ must be $L$-Lipschitz w.r.t. $\theta$. Can this property be directly inferred from existing assumptions? If not, this constitutes an additional assumption, and the authors should justify its reasonableness. Additionally, does $L$ depend on $|S|$, $|A|$ or $t_{\rm mix}$?

2. What step-size is used in Theorem 1? Is it a constant or decaying step size?  Furthermore, how does the step size scale with $T$ and the mixing time? This is a crucial aspect and should be explicitly stated in or before the theorem.

3. What are the definitions of $\Lambda_p$ and $\Lambda_q$ in Eq.(35)? Can they take arbitrary values?

4. The proposed algorithm operates on a two-timescale framework, as evident from Algorithm 1, where multiple critic updates (update on $\zeta$ in line 12) occur before a single actor update (update on $\theta$ in line 27). Two-timescale methods are generally easier to analyze since the outer loop can proceed after the inner loop converges. However, prior works such as NAC-CFA and MAC already focus on single-timescale algorithm, which are more commonly used in practice. In the comparison table, I suggest adding a column indicating whether each method follows a single- or two-timescale approach.

5. Given the previous point, the novelty of this work may be questionable unless the authors can demonstrate that the sample complexity analysis remains valid for a single-timescale version of Algorithm 1.

**Relation To Broader Scientific Literature:**

N/A

**Theoretical Claims:**

Yes

---

> ### Author Rebuttal · Authors · 2025-03-31
>
> **Markovian Noise**: We agree that applying projections or invoking the assumption of finite action space is common in the literature. For example, the MAC algorithm derives bounds using MLMC estimates from (Dorfman and Levy 2022), which assume a finite state space and a bounded domain, thereby necessitating a projection step in the critic update. Our analysis models the NPG and critic updates as stochastic linear recursions with Markovian noise, which we show to converge without invoking the above preconditions. A similar result can be found in (Beznosikov et al. 2023). However, their update differs from ours since we deal with biased estimates, which are not considered in the cited paper.
>
> **(Q1) Lipschitz Property**: Lipschitz continuity and smoothness of the function $J$ can be easily proven within the framework of discounted-reward MDPs (Lemma 2, Mondal and Aggarwal 2024). In this case, $L$ depends only on the discount factor. However, to the best of our knowledge, no such result is known in the average-reward case. We want to emphasize that nearly *all* papers analyzing general parameterized policy gradient methods for average-reward MDPs assume the stated properties of the $J$ function. These properties are expected to hold in practice since they ensure that a slight change in the parameter does not result in a large change in the $J$ function. In our assumption, $L$ is a constant that does not depend on other model-specific parameters.
>
> **(Q2) Step Sizes**: The values of the step sizes $\beta$ and $\gamma$ are given in Theorems 3 and 4, respectively: $\beta = \frac{4\log H}{\lambda H}$, and $\gamma = \frac{2\log H}{\mu H}$. Moreover, the value of $\alpha$ is mentioned at the top of Page 20, line 1045: $\alpha = \frac{\mu^2}{4G_1^2L}$. Observe that none of these parameters depend on the mixing time, $t_{\mathrm{mix}}$, which is consistent with our claim that our algorithm does not require the knowledge of $t_{\mathrm{mix}}$. In the revised version, we will explicitly state the values of these parameters in Theorem 1.
>
>  **(Q3) Regarding $\Lambda_P$ and $\Lambda_Q$**: The terms $\Lambda_P$ and $\Lambda_q$ in (35) are parts of Theorem 2 that analyzes the convergence of a general stochastic linear recursion (SLR). Specifically, Eq. (35) forms a precondition for Theorem 2. We recognize that both NPG and critic updates can be formulated as SLRs. Moreover, these updates satisfy a precondition similar to (35). In particular, (53) and (54) allow us to choose $\Lambda_P=c_\beta+3$ and $\Lambda_q=c_\beta+1$ for the critic update while (55) dictates $\Lambda_P = G_1^2$ and $\Lambda_q = \mathcal{O}(G_1t_{\mathrm{mix}})$ for the NPG update. Note that, although $\Lambda_q$ for the NPG update is dependent on $t_{\mathrm{mix}}$, it is not used as an input to our algorithm, thereby maintaining the claim that our algorithm does not require knowledge of the mixing time.
>
> **(Q4) On Two-Timescale Approach**: We emphasize that no algorithm (be it single- or two-timescale) exists in the literature that achieves the optimal convergence rate of $\mathcal{O}(1/\sqrt{T})$ for average-reward MDPs with general parameterized policies via Markovian sampling. Our natural actor-critic-based algorithm achieves this goal, and that itself is a contribution worth highlighting.
>
> Secondly, the algorithms mentioned by the reviewer are not single timescale. We note that the paper NAC-CFA (https://arxiv.org/pdf/2406.01762) mentions (page 5) that ``Here, the AC and NAC algorithms in Algorithm 1 are single-loop, single sample trajectory and two timescale." Similarly, the MAC algorithm is also two timescale since the critic loop has an order-larger step size which makes the algorithm two timescale. Thus, we note that both the mentioned algorithms are two timescale. We will mention this in the paper and enlist designing a one-scale algorithm as a future work.
>
> **(Q5) Novelty**: Our main contribution is designing an algorithm for average-reward MDPs that achieves the optimal $\mathcal{O}(1/\sqrt{T})$ rate for general parameterized policies with Markovian sampling. This is a contribution worth highlighting because no algorithm, be it single- or two-timescale, achieves this feat. Moreover, as previously pointed out, all known actor-critic algorithms on average-reward MDPs with general parameterized policies are all, to the best of our knowledge, two-timescale. In summary, we respectfully disagree with the reviewer's stance that not being a single-timescale algorithm is a good enough reason to discount the merit of our work.
>
> The optimization literature teaches us that single-timescale algorithms typically develop based on the intuitions provided by two-timescale algorithms. We are, therefore, hopeful that the intuitions provided by our algorithm will pave the way to designing the first single-timescale actor-critic algorithm for average reward MDPs with general parameterized policies. We will mention this as a future work in the revised paper.

---

> > ### Comment · Reviewer_en8p · 2025-04-02
> >
> > Thank you for the authors’ response. The clarifications on Markovian noise, the Lipschitz and smoothness property, and the definitions of $\Lambda_P, \Lambda_Q$ make sense to me. I encourage the authors to explicitly include the clarification on the Lipschitz and smoothness properties in the paper. Additionally, I have noticed a slight difference in previous works regarding this assumption, as PHPG assumes the smoothness of $\nabla J$ rather than $J$ itself. I believe this distinction is worth highlighting in the paper as well.
> >
> > Regarding the distinction between single- and two-timescale algorithms, I apologize for the confusion in my initial comment. My intent was to distinguish between single-loop and double-loop algorithms rather than single- and two-timescale methods. It is evident that the algorithm proposed in this paper follows a single-loop structure, as multiple critic updates occur before a single actor update. However, I believe that comparing the analysis of single-loop and double-loop algorithms may not be entirely fair, as in double-loop algorithms, the outer loop proceeds only after the inner loop has converged, which simplifies the theoretical analysis. This distinction should be explicitly noted in the comparison table for clarity.

---

> > > ### Author Response · Authors · 2025-04-03
> > >
> > > Thank you for the clarification. We will definitely incorporate the suggested discussion on the smoothness assumption in the revised paper. Regarding the algorithmic novelty, our point still stands. There exists no algorithm in the literature (be it single- or double-loop) that achieves the order-optimal global convergence rate with general parameterized policies via Markovian sampling. Our proposed algorithm accomplishes this feat, making it a significant contribution worth highlighting.
> > >
> > > Additionally, the paper https://openreview.net/pdf?id=jh3UNSQK0l on single-timescale Actor-Critic (AC) highlights that single-loop two-timescale AC still shares the same drawbacks as double-loop AC. Specifically, even with single loop, the paper notes that these may be inefficient in practice (“However, it is still considered inefficient as the actor update is artificially slowed down") and relies on a decoupled analysis of the policy and critic updates (“ The two-timescale allows the critic to approximate the desired Q-value in an asymptotic way, which enables a decoupled convergence analysis of the actor and the critic”). That said, we acknowledge that analyzing a double-loop structure might be easier than its single-loop counterpart. We will include this point in the paper and outline the design of a single-loop algorithm that achieves the optimal convergence rate as a direction for future work.
> > >
> > > We hope our rebuttal has adequately addressed all of your concerns. Given these clarifications, we would greatly appreciate it if you could reconsider your evaluation and adjust the score accordingly to reflect your updated perception of the paper.

---

### Official Review · Reviewer_nDZw · 2025-03-07

**Overall Recommendation:** 3

**Summary:**

The paper establishes $O(\epsilon^{-2})$ sample complexity for MLMC-NAC algorithm, claims to significantly improve existitng result of $O(\epsilon^{-4})$.

State-space independent result is not surpring as NAC (in exact gradient NAC case) is also state-independant.

**Claims And Evidence:**

Yes, seems so.

**Essential References Not Discussed:**

The paper doesn't cite  [2], establishing 1/T convergence rate for average reward case (for exact gradient case).

This recent work [1] also addresses the convergence of dynamic programming techniques in robust average reward MDPs and thus should be included in the related works literature."



[1] @inproceedings{murthy2023modified,
  title={Modified policy iteration for exponential cost risk sensitive mdps},
  author={Murthy, Yashaswini and Moharrami, Mehrdad and Srikant, R},
  booktitle={Learning for Dynamics and Control Conference},
  pages={395--406},
  year={2023},
  organization={PMLR}
}

[2] @inproceedings{
kumar2025global,
title={Global Convergence of Policy Gradient in Average Reward {MDP}s},
author={Navdeep Kumar and Yashaswini Murthy and Itai Shufaro and Kfir Yehuda Levy and R. Srikant and Shie Mannor},
booktitle={The Thirteenth International Conference on Learning Representations},
year={2025},
url={https://openreview.net/forum?id=2PRpcmJecX}
}

**Experimental Designs Or Analyses:**

No.

**Methods And Evaluation Criteria:**

Yes, seems so.

**Other Comments Or Suggestions:**

Suggestion: Presentation can be improved.

**Other Strengths And Weaknesses:**

Strengths:  Good theoretical result.

**Questions For Authors:**

Question: How this MLMC NAC algorithm can be used in large scale problems.  Does this paper has any message for the practioners?

**Relation To Broader Scientific Literature:**

In my personal opinion, this NAC algorithm guanrantees are of theoretical and conceptual interests. I am not sure, this flavour of NAC algorithms are used in large scale problems.

**Theoretical Claims:**

I didn't verify the proofs.

---

> ### Author Rebuttal · Authors · 2025-03-31
>
> **(a) Related Works**: Thanks for mentioning references [1] and [2]. We will include them in the revised version of our paper. In addition to [1], we will also include some other works on robust MDPs.
>
>
> **(b) Message for Practitioners**: Our algorithm is designed for large state and action spaces, with parameterized representations for both the actor and critic, making it well-suited for large-scale problems. While there are multiple practical insights, we highlight a few key ones below.
>
> 1. **Sampling**: Many existing policy gradient-type algorithms assume access to a simulator capable of generating independent trajectories at will from a given state distribution (e.g., see [1], [2]). Typically called the i.i.d sampling, the procedure of generating independent and identically distributed trajectories greatly simplifies the algorithm design and analysis. However, for many practical applications, building an accurate simulator is itself a difficult problem. Fortunately, our algorithm works on a single trajectory, eliminating the need for a simulator.
>
>
> 2. **Dependence on Unknown Parameters**: As stated in the paper, many previous works assume knowledge of mixing time and hitting time [3], [4], which are difficult to obtain for most applications. Our algorithm does not require the knowledge of the above parameters, thereby making it easier to adopt in practice.
>
> 3. **Memory/Space Complexity**: Our algorithm is memory-efficient. One can verify that the memory complexity of our algorithm is $\mathcal{O}(\max\\{\mathrm{d}, m\\})$ where $\mathrm{d}, m$ are the sizes of the policy parameter, $\theta$, and the critic parameter, $\zeta$ respectively. It is to be noted that although the Fisher matrix, $F(\theta)$ (and its estimates) require $\mathcal{O}(\mathrm{d}^2)$ space, one only needs to store quantities of the form $F(\theta)\omega$, $\omega\in\mathbb{R}^{\mathrm{d}}$, which need $\mathcal{O}(\mathrm{d})$ space.
>
> 4. **Computational Complexity**: One way of computing the natural policy gradient (NPG) $\omega_\theta^* = [F(\theta)]^{-1}\nabla_{\theta}J(\theta)$ is first obtaining an estimate of the Fisher matrix $F(\theta)$, and then directly using its pseudo-inverse to compute $\omega_{\theta}^*$. Such a method was adopted in [5]. We, instead, pose the problem of computing the NPG as a stochastic least square problem with Markovian noise. This eliminates the computationally expensive process of inverting a regularized Fisher matrix estimate (whose computational complexity is $\mathcal{O}(\mathrm{d}^3)$ where $\mathrm{d}$ denotes the size of the policy parameter). It can be checked that the computational complexity of our algorithm for a given iteration instance $(k, h)$ is $\mathcal{O}(\mathrm{d}^2+m^2)$ where $m$ is the size of the critic parameter.
>
> 5. **Convergence Rate**: Finally, the convergence rate of our algorithm is optimal in the horizon length, $T$. Practically, it indicates that our algorithm takes a relatively small number of training iterations to reach a certain accuracy.
>
> [1] Liu, Y., et al., An improved
> analysis of (variance-reduced) policy gradient and natural
> policy gradient methods. Advances in Neural Information
> Processing Systems, 33:7624–7636, 2020.
>
> [2] Mondal, W. U. and Aggarwal, V. Improved sample complexity analysis of natural policy gradient algorithm with general parameterization for infinite horizon discounted reward Markov decision processes. International Conference on Artificial Intelligence and Statistics, 2024.
>
> [3] Bai, Q., et al., Regret analysis of policy gradient algorithm for infinite horizon average reward Markov decision processes. Proceedings of the AAAI Conference on Artificial Intelligence, 2024.
>
> [4] Ganesh, S. et al., Order-Optimal Regret with Novel Policy Gradient Approaches in Infinite Horizon Average Reward MDPs. In The 28th International Conference on Artificial Intelligence and Statistics, 2025.
>
> [5] Xu, T., et al., Improving sample complexity bounds for (natural) actor-critic algorithms. Advances in Neural Information Processing Systems, 2020.

---

### Official Review · Reviewer_8Gaz · 2025-03-11

**Overall Recommendation:** 2

**Summary:**

This paper studies the convergence of Actor-Critic algorithm in average reward setting of reinforcement learning. The paper proposed a multi-level Monte-Carlo based natural actor critic algorithm which achieves a $\tilde{O}(1/\sqrt{T})$ global convergence rate to a neighborhood of the optimal policy, where $T$ is the horizon length of the sampled trajectory.

### Update after rebuttal: I appreciate the authors’ effort in rebuttal. Although some of my concerns have been addressed, the remaining concern is with regard to the trajectory length. Throughout the paper and most of the rebuttal it seems to be $\max\(2^{Q_{kh}}, T_{\max}\)$. However, it was pointed out in the last reply to rebuttal comment that it was a typo. Various places (line 6, 11, 23) of algorithm 1 in the paper used max operator rather than min operator. It appears the paper can benefit from greater clarity in conveying the idea.

**Claims And Evidence:**

Yes.

**Essential References Not Discussed:**

The work lacks the discussion of a very important branch of sample efficiency literature for example A and references within. Instead of convergence rate, which seems to only measure critic estimation and NPG estimation, the sample complexity is a more practical measure of complexity of as it represents the true resource consumption from the sample necessity. In the context of the current paper, the paper should include the policy update component.

[A] Xu T, Wang Z, Liang Y. Improving sample complexity bounds for (natural) actor-critic algorithms[J]. Advances in Neural Information Processing Systems, 2020, 33: 4358-4369.

**Experimental Designs Or Analyses:**

NA

**Methods And Evaluation Criteria:**

Proposed method makes sense. However, there’s no empirical evaluations for the current version.

**Other Comments Or Suggestions:**

Potential typo: Line 320, left column, left hand side of the inequality, subscript $d^{\pi*}$ should be $\nu^{\pi*}$?

**Other Strengths And Weaknesses:**

Strength: The paper studied the challenging average reward setting with finite-time convergence result. The analysis for this setting is in general non-trivial.

Weakness:

Sample complexity perspective: it seems to measure the sample trajectory length for each pair of $k, h$ values. But in fact, the algorithm relies on $KH$ number of trajectories for the entire NAC algorithm to work. Based on Theorem 1, $KH = O(T)$ is large as well. This potentially poses sample inefficiency. Please elaborate or correct if the reviewer misunderstood.

**Questions For Authors:**

Please see the weakness point.

**Relation To Broader Scientific Literature:**

The key contributions relate to the convergence performance of actor-critic algorithmic framework in RL, especially in the average reward setting. The paper proposed the model-free approach with order-wise comparable convergence result as the model-based approach.

**Theoretical Claims:**

The reviewer didn’t check the correctness of the proofs.

---

> ### Author Rebuttal · Authors · 2025-03-31
>
> **(a) Sample Complexity**: We note that there is a misunderstanding here. The convergence rate and the sample complexity are two faces of the same coin, and one can be derived from the other, as long as the error metric is taken to be the same. For example, consider our algorithm that uses a trajectory of length $\max\\{2^{Q_{kh}}, T_{\max}\\}$ where $Q_{kh}\sim \mathrm{Geo}(1/2)$ for each $k, h$. It is easy to check that $$\mathbf{E}[\max\\{2^{Q_{kh}}, T_{\max}\\}]\leq \mathcal{O}(\log T_{\max})$$
>
> Since we have taken the number of iterations of $k, h$ to be $K=\Theta(\sqrt{T})$ and $H=\tilde{\Theta}(\sqrt{T})$, and $T_{\max} = H^2 = \tilde{\Theta}(T)$, the expected number of state transition samples used by our algorithm is $\mathcal{O}(KH\log T_{\max}) = \tilde{\Theta}(T)$. Theorem 1 exhibits that for such choices of parameters, our algorithm achieves $\tilde{\mathcal{O}}(1/\sqrt{T})$ global error (up to some additive factors of $\epsilon_{\mathrm{bias}}$ and $\epsilon_{\mathrm{app}}$). This establishes the convergence rate of our algorithm. Alternatively, if we want to ensure a global error of $\epsilon$ (up to the additive factors stated before), the expected number of state transition samples required would be $\tilde{\mathcal{O}}(\epsilon^{-2})$. This expresses the same result in terms of sample complexity.
>
> Although both notions are the same, the literature on average-reward MDP typically expresses the results in terms of the convergence rate, while the literature on discounted-reward MDP  typically adopts the sample complexity framework. The cited paper [A] analyzes a discounted-reward setup where the common metric is the sample complexity. The same article has also been cited in our paper (line 118), where we mention its global convergence rate to be $\mathcal{O}(T^{-1/3})$. This is obtained from their equivalent sample complexity result of $\mathcal{O}(\epsilon^{-3})$. It is to be noted that even in the discounted MDP setup, there is no actor-critic algorithm that achieves $\mathcal{O}(\epsilon^{-2})$ sample complexity for general parameterization.
>
>
> **(b) Typo**: Thanks for pointing out the typo. We will correct it in the revised version.

---

> > ### Comment · Reviewer_8Gaz · 2025-04-08
> >
> > I would like to thank the authors for the response.
> >
> > 1) However, the order-wise result in the response should be $$\mathbf{E}[\max \(2^{Q_{kh}},T_{\max} \)] \ge T_{\max} = \Theta(T_{\max})$$ instead of the stated result.
> >
> > 2) In the following literature [A], the sample complexity result is $\tilde{\mathcal{O}}(\epsilon^{-2})$ for discounted setting.
> >
> > [A] Xu, Tengyu, Zhe Wang, and Yingbin Liang. "Improving sample complexity bounds for (natural) actor-critic algorithms." Advances in Neural Information Processing Systems 33 (2020): 4358-4369.
> >
> > I would like to maintain the score.

---

> > > ### Author Response · Authors · 2025-04-08
> > >
> > > 1. We apologize for the typo. The length of the trajectory should be $\min\\{2^{Q_{kh}}, T_{\mathrm{max}}\\}$. In this case, one can see that
> > > $$
> > > \begin{align}
> > > \mathbf{E}\left[\min\\{2^{Q_{kh}}, T_{\max}\\}\right] &= \sum_{q=1}^{\lfloor \log_2 T_{\max}\rfloor} 2^q \mathbf{Pr}(Q_{kh}=q)+ \sum_{q=\lfloor \log_2 T_{\max} \rfloor + 1}^{\infty} T_{\max} \mathbf{Pr}(Q_{kh}=q) \\\\
> > > &= \sum_{q=1}^{\lfloor \log_2 T_{\max}\rfloor} 2^q\times 2^{-q} + \sum_{q=\lfloor \log_2 T_{\max} \rfloor + 1}^{\infty} T_{\max}2^{-q} \\\\
> > > &= \lfloor \log_2 T_{\max}\rfloor + T_{\max} 2^{-\lfloor \log_2 T_{\max}\rfloor} \leq \lfloor \log_2 T_{\max}\rfloor +2 = \mathcal{O}\left(\log_2 T_{\max}\right)
> > > \end{align}
> > > $$
> > >
> > > Therefore, our conclusions remain unchanged.
> > >
> > > 2. We note that the cited paper [A] has an incorrect sample complexity for NAC which was subsequently corrected in the arXiv version (https://arxiv.org/pdf/2004.12956).  In this paper, the sample complexity $\mathcal{O}(\epsilon^{-2})$ is established via the vanilla actor-critic (AC) algorithm is to ensure a first-order *local* or *stationary* convergence error of $\epsilon$ (please see the footnotes following the comparison table in the mentioned paper). In contrast, the same paper establishes a sample complexity of $\mathcal{O}(\epsilon^{-3})$ via a natural actor-critic (NAC) algorithm to ensure an $\epsilon$ *global* error. We reported the global convergence result in our paper.
> > >
> > > We hope the above clarification resolves all of your remaining concerns.

---

### Official Review · Reviewer_EoWh · 2025-03-12

**Overall Recommendation:** 3

**Summary:**

This work considers the average-reward RL setting with general policy parametrization. The authors improve over the state-of-the-art global convergence guarantee from a rate of $T^{-1/4}$ to $T^{-1/2}$ without requiring knowledge of the mixing or hitting times. This is done by adopting a multi-level MC procedure to estimate the relevant quantities and through a tighter analysis of the derived estimates.

**Claims And Evidence:**

The claims made in the submission are clear and supported by theoretical results.

Concerning the table, the authors states that their approach works for infinite state and actions, and similar claims are made for some of the related works. However, by inspecting those works, it appears that their results are defined with respect to large but finite spaces. Can the authors comment on this?

**Essential References Not Discussed:**

In my opinion, the related literature has been thoroughly discussed.

**Experimental Designs Or Analyses:**

No experiments.

**Methods And Evaluation Criteria:**

No experiments were presented in the work.

**Other Comments Or Suggestions:**

See above

**Other Strengths And Weaknesses:**

Among the strengths of the work, I mention the newly achieved result in terms of convergence for the considered setting without knowledge of the mixing time and with infinite state and actions.

Concerning the weaknesses, the authors do not properly highlight the technical novelty in terms of theoretical analysis. Indeed, this work shares many similarities both in terms of assumptions and in terms of methods with the one of Patel et al. (2024). I believe that the work may benefit from a clearer comparison between the current work and the one just mentioned.

Another weakness is the absence of numerical simulations.

**Questions For Authors:**

See Sections above

**Relation To Broader Scientific Literature:**

The paper presents an improvement in terms of global convergence guarantees with respect to state-of-the-art approaches. A similar convergence result was achieved in Ganesh et al. (2024) but in a slightly different setting with finite state and actions and knowledge of the mixing time.
A work closer in terms of assumptions and setting is the one from Patel et al. (2024) but achieves a worse convergence rate of $T^{-1/4}$.

**Theoretical Claims:**

I checked the proof outline in Section 5 which seems correct to me.

---

> ### Author Rebuttal · Authors · 2025-03-31
>
> > Concerning the table, the authors state that their approach works for infinite states and actions, and similar claims are made for some of the related works. However, by inspecting those works, it appears that their results are defined with respect to large but finite spaces. Can the authors comment on this?
>
> We agree that some related works (NAC-CFA and MAC) assume large but finite state spaces, and we have mistakenly given them more credit by stating that their conclusions extend to infinite state spaces. We will make changes in the revised paper accordingly. In this context, we want to emphasize that our work does apply to infinite states and action spaces, and there is no mistake in that claim.
>
>
> > Concerning the weaknesses, the authors do not properly highlight the technical novelty in terms of theoretical analysis. Indeed, this work shares many similarities both in terms of assumptions and in terms of methods with the one of Patel et al. (2024). I believe that the work may benefit from a clearer comparison between the current work and the one just mentioned.
>
>
> The assumptions used in this work are commonly used in the literature and thus are similar to those used in multiple works including (Patel et al., 2024).
>
>  Although our methods bear *some* similarities with that of (Patel et al., 2024), there are multiple novel components in the analysis that allow us to establish an order-optimal convergence rate of $\tilde{O}(T^{-1/2})$ for average reward MDPs without the knowledge of mixing time, a feat achieved by no other work in the literature. While existing AC analyses (including that in the cited paper) apply relatively loose bounds, we refine the analysis to get sharper results. The first step towards this goal is to establish that the global convergence error is bounded by terms proportional to the bias and second-order error in the NPG estimation (Lemma 1). In prior AC works (including the cited paper), a coarser bound was used instead of the NPG bias that led to an error of the form $\mathbb{E} ||\xi_t - \xi^\star||$ in the global convergence bound where $\xi_t$ is the critic estimate at time $t$ and $\xi^\star$ denotes the true value. Utilizing Lemma 1 and Theorem 3, our analysis refines this term to $||\mathbb{E}[\xi_t] - \xi^\star ||$, which can be significantly smaller than the previous estimate. It is to be noted that bounding $||\mathbb{E}[\xi_t] - \xi^\star||$ remains challenging due to Markovian sampling.  Interestingly, we observe that both critic and NPG updates can be interpreted as linear recursions with Markovian noise.  Efficient analysis of such a linear recursion is another novelty of our paper. In Theorem 2, we obtain a convergence rate for a generic stochastic linear recursion. This, along with the improved error estimates, form a basis for Theorems 3 and 4, which finally leads to our desired result in Theorem 1.
>
> In summary, improving the error estimates, recognizing the NPG and critic updates as linear recursions, and performing a sharp convergence analysis of a general stochastic linear recursion are the novel cornerstones of our analysis.
>
> >Another weakness is the absence of numerical simulations.
>
> The focus of the paper is obtaining the first theoretical result for a state-action space size independent global convergence rate of $\tilde{\mathcal{O}}\left(T^{-1/2}\right)$ for general parameterized policies in average reward infinite-horizon MDPs, using a practical algorithm that does not require the knowledge of mixing times. Evaluation of this work on some practical applications is left as future work.

---

### Decision · Program_Chairs · 2025-05-01

**Decision:**

Accept (poster)

**Comment:**

This manuscript identifies the main issue of the previous analysis on natural policy gradient via actor-critic approach, and proposes new technical tools that view both the critic update and natural policy gradient finding as a stochastic linear recursion, to overcome the issue and obtain minimax optimal convergence rate in terms of $T$. Although most of the theoretical analysis does not deviate away from the existing work, most of the reviewers do find that the proposed technical improvement is solid.

As multiple reviewers pointed out, the presentation should be further improved. The authors should highlight the key component of the technical improvements that can let the reader easily separate the manuscript with other existing works. There are also some other typos which make one of the reviewer confused about the total sample complexity. I do hope the authors can make the necessary adjustments based on reviewers' comments.